# Research on China's Manufacturing Industry Moving towards the Middle and High-End of the GVC Driven by Digital Economy

Rongrong Zhou [1], Decai Tang [1,2,*] , Dan Da [3,*], Wenya Chen [2], Lin Kong [1] and Valentina Boamah [2]

1     School of Law and Business, Sanjiang University, Nanjing 210012, China;
      zhou_rongrong002338@sju.edu.cn (R.Z.); kong_lin@sju.edu.cn (L.K.)
2     School of Management Science and Engineering, Nanjing University of Information Science & Technology,
      Nanjing 210044, China; 20201242001@nuist.edu.cn (W.C.); 20215242005@nuist.edu.cn (V.B.)
3     School of Business, Jiangsu Open University, Nanjing 210000, China
*     Correspondence: tangdecai@nuist.edu.cn (D.T.); dadan@jsou.edu.cn (D.D.)

**Abstract:** A country's manufacturing industry is often an important route for national prosperity, but it is also a conduit by which a digital economy may become truly useful. This is so the deep integration of the digital economy and manufacturing industry can enhance independent innovation efficiencies, promote the development of advanced manufacturing clusters, and constantly spawn new models, forms of business, and industries. Consequently, it is crucial to improve China's global manufacturing value chain. This article starts with an analysis of the development status and competitiveness of the digital economy in China and abroad. It establishes a structural equation and uses the latest data from the World Input-Output and Asian Development Bank databases. It introduces new variables, such as digitization, research and development (R&D) investment, and industrial scale to empirically analyze China's manufacturing industry's global value chain (GVC). The results show that the digitization of China's manufacturing industry can increase the forward participation of GVC in the manufacturing industry to improve the division status of GVCs. Analyses suggest that due to insufficient R&D investment in the division of labor in the GVCs, China's manufacturing industry is prone to low-end lock-in, inefficient industrial structures, and weak innovation ability. Consequently, the following suggestions are proposed: China's manufacturing industry needs to accelerate digital transformation, increase R&D investment, actively participate in the division of labor in the GVCs, and enhance core competitiveness.

**Keywords:** digitization; manufacturing value chain in China; promotion countermeasure





## 1. Introduction

Affected by the COVID-19, the supply and demand of the global value chain have been squeezed. As a result, the production and supply of many products are interrupted, and developed countries pay more attention to the security of the global supply chain. Developed countries have begun to strengthen the layout of domestic production capacity, and the manufacturing supply chain tends to shrink and shorten to meet the needs of national strategic security [1].

At the same time, a new round of scientific and technological revolutions has promoted the digital economy's rapid development. Digital technologies, such as the Internet of things, big data, and cloud computing, are widely used to promote cross-border integration and innovation in different industries effectively. The integrated development of the digital economy and manufacturing industry will improve the innovation level of the manufacturing industry. This will continue to give birth to new models, new forms of business, and new industries in the manufacturing industry [2]. In this way, the utilization efficiency of production factors, such as human and material resources in the manufacturing industry,

can be improved, the waste of resources can be reduced, and the sustainable development of the manufacturing industry can be realized. Finally, the global manufacturing industry's industrial chain, supply chain, and value chain are subversive layouts and changes. As a major manufacturing country, China's manufacturing global value chain division status has also changed, facing opportunities and challenges. Therefore, accelerating the development of the digital economy is of great significance in enhancing the division of labor status of China's manufacturing global value chains (GVCs) [3].

Since 2010, the added value of China's manufacturing industry has ranked first in the world for 12 consecutive years. In 2021, the added value of China's manufacturing industry was USD 4.86 trillion, accounting for 27.4% of China's GDP. The digital economy has also been accelerating [4]. According to the statistics of China's Ministry of industry and information technology, the scale of the added value of China's digital economy has increased from CNY 18.6 trillion in 2015 to CNY 39.2 trillion in 2020 [5]. Its proportion in GDP also increased from 27.5% to 38.6%, with a compound annual growth rate of 20.6% [6]. Although the scale of China's manufacturing industry ranks first in the world, its participation and division of labor in manufacturing GVC still lags behind developed countries. This is because the technological level of China's manufacturing industry is low and lacks core competitiveness. The GVCs' division of labor is located in the processing link with low added value. Developed countries occupy a high position in the GVCs by providing core components to other countries. This makes China's manufacturing products easier to be replaced and causes weak international competitiveness. China urgently needs to realize the transformation and upgrading of the manufacturing industry and improve the division of labor position of the Chinese manufacturing industry in GVCs [7].

This article focuses on the promotion of GVC in China's manufacturing industry by the digital economy from the following aspects. Firstly, the input–output method is used to compare and analyze the manufacturing digitization level of 18 key countries with the manufacturing digitization index (DMI) [8]. Secondly, based on the input and output table of WIOD, the GVC division status of the manufacturing industry in 18 key countries is compared and analyzed by using the decomposition method of export trade added value [9,10]. Thirdly, the fixed effects model is established to analyze the impact of manufacturing digitization on GVC. Finally, according to the empirical analysis results and the development status of China's digital economy, this article puts forward the path and countermeasures to promote the rise of GVC in China's manufacturing industry.

The existing research on the GVCs' division of labor status is mainly at the macro analysis level. Although some research achievements have achieved breakthroughs, they are still limited to a certain country, which is not fully in line with the actual situation of GVC. The research on the impact of the digital economy on GVCs in the manufacturing industry is still at the level of theoretical mechanism research, and there is a lack of a quantitative analysis evaluation model. Therefore, this article makes up for the relevant gap in the previous research.

The contribution of this article is mainly reflected in the following two aspects. Firstly, using the value-added decomposition method of export trade, the GVCs' division of labor status in the manufacturing industry in China and 17 key countries from 2000 to 2017 was measured. Through the comparative analysis of 18 countries, this article points out the key problems of GVC in China's manufacturing industry. It also makes up for the lack of comparison between China's GVC division of labor and other countries in the existing literature research. Thus, this article completely and accurately describes the division of labor status of China's manufacturing industry and various departments of the manufacturing industry in GVC. Secondly, it establishes a fixed effect regression model to quantitatively analyze the impact of the digital economy on improving the overall and divisional GVC division status of China's manufacturing industry. It makes up for the lack of quantitative analysis in the current research on the promotion of GVC in the manufacturing industry by the digital economy. Based on the perspective of the data economy, this article puts

forward the path of GVC upgrading of China's manufacturing industry and the measures for China's manufacturing industry to participate in GVC governance.

## 2. Literature Review

### 2.1. Digital Economy and Global Value Chain Theory

The concept of the digital economy was originally proposed by Tapscott [11], which refers to the digital revolution of information technology and a new economy based on the networking of human intelligence. Later, Negroponte defined the digital economy as "an economic form that uses bit technology instead of atoms", and its essential characteristic is an economic form based on the Internet; it mainly includes industrial digitization and digital industrialization [12,13]. The definition of the digital economy is commonly segmented into three: core, narrow, and broad definitions. The core definition is that the digital economy is the core sector of digital economic activities, namely the ICT industry [9]. The narrow definition identifies the digital economy as an economic activity that utilizes numbers, that is, a digital sector dominated by digital products and services. The broad definition holds that the digital economy is the economic effect of digital-driven industrial upgrading. That is, it includes the core industries of the digital economy, such as the ICT industry, and the contribution of the ICT industry to agriculture and service industries [14].

The digital economy was also defined as "the technical process of converting analogue signals into a digital form, and ultimately into binary digits, and is the core idea brought forward by computer scientists since the inception of the first computers" based on work by Chen [15] and Hess [16]. In other words, digitization implies the technical potential of separating information from physical data carriers and storage. On the other hand, digitalization is "the manifold sociotechnical phenomena and processes of adopting and using these (digital) technologies in a broader individual, organizational, and societal context". This definition aligns with the statement of Yoo et al. [17,18]; the digital economy consists of both social and technical dimensions. The fourth industrial revolution is driven by real-time data exchange and flexible manufacturing, underpinned by the advancement of ICT and data storage, thus enabling customized production [19].

As mentioned earlier, Industry 4.0 can be understood through its fundamental components, the eight key enabling technologies. In other words, these digital technologies are the technologies that enable digitalization in the fourth industrial revolution [20].

Regarding the definition of GVC theory, Porter [18] believes that "value creation activities are divided into basic activities (including production, marketing, transportation, after-sales service, etc.) and support activities (including raw material supply, technology, human resources, and finance, etc.), which connected in the process of company value creation, formed a value chain within the company's value creation behavior, and related to the value chains of other enterprises within a value system composed of many value chains [21]. Gereffi [22] defines GVC upgrading as an enterprise, country, or region that ultimately obtains benefits from global production, such as safety, profit, added value, etc., through the development of higher value-added industrial activities. The dynamic upgrading of the GVC is closely related to economic activities, such as production and export, mainly including the processing of imported raw materials, original equipment manufacturing (OEM), original design manufacturing (ODM), and original brand manufacturing (OBM). Gereffi proposed the value chain upgrade path of enterprises from processing and manufacturing to original equipment manufacturing (OEM) and then to original design manufacturing (ODM) and original brand manufacturing (OBM). As the main body of the international production network, multinational corporations closely link various production-related enterprises worldwide into the global production chain of commodities. The key to the global commodity chain is the node, and any node contains raw material input, operation organization, marketing, and other content links [21]. The GVC mainly studies the relationship between global economic organization and division of production based on the vertical dimension of value. Humphrey and Schmitz proposed

four modes of industrial upgrading in GVCs: process upgrading, product upgrading, function upgrading, and chain upgrading [23].

*2.2. Theoretical Mechanism of the Division of Labor and Level of Participation in Global Value Chains*

Import and export trade is one of the core driving forces of the "troika" (investment, consumption, import and export trade) driving economic growth [24]. From the perspective of global economic growth in the past 40 years, there is a big difference between countries with higher forward and backward participation in GVCs based on comparative advantages. Sampatha and Vallejob [25] found that countries that mainly trade in intermediate goods have a higher degree of forwarding participation in the division of labor in the GVC. Their products have a higher level of export technology and added value. They tend to have greater discourse power, that is, pricing power. Countries with a higher degree of backward participation have lower technical content and added value of exported final products—these countries find it difficult to be unshackled from factor-driven trade.

Scholars have studied the factors affecting the division of labor in GVCs from various perspectives. Humphrey believes that developing countries can upgrade their industrial value chains through the opening, active innovation, and learning by embedding the GVC division system dominated by developed economies such as Europe, America, and Japan [2,26]. Koopman [27], Gereffic used share of vertical specialization (VSS) to measure the status of the international division of labor and found that cutting tariffs and transportation costs would expand the level of vertical specialization, thereby promoting trade growth and improving the status of GVCs [28].

Scholars believe that the production link between countries no longer occurs after the final production of the product but occurs at various stages of product research and development, manufacturing, marketing, and operation management [29]. In deepening the division of labor in the value chain, some countries have seized the opportunity of global industrial restructuring and integrated it into the global production network system by relying on their comparative advantages in a certain production link in the product chains. Countries participating in global production have gradually realized the upgrading of their domestic industries and the promotion of the division of labor in the GVC. Especially for developing countries, participating in the international division of labor in the value chain can promote employment in the short term and improve economic benefits [22]. More importantly, in the international division of labor and cooperation process, through the technology spillover effect and learning effect, the quality and technology of domestic export products are promoted, and the division of labor in the value chain is improved [8]. Elements such as technology and capital are the key driving forces for improving the division of labor in the GVC of high-tech industries [30].

Developed countries participate in the international division of labor through multinational corporations, promote the flow of capital, technology, and human resources, and organize the production and sales of goods worldwide. The GVCs can be improved by optimizing the allocation of resources [31]. Based on the advantages of the low cost of domestic production factors and convenient transportation, developing countries participate in the international division of labor to improve the economic benefits of enterprises. In addition, they promote the improvement of the quality and technology of domestic export products through the technology spillover effect and learning effect. Furthermore, they realize the transformation and upgrading of domestic industries and gradually improve the division of labor status of GVCs [32].

Gereffi and others believe that GVC governance refers to the management behavior of the value chain leaders to coordinate and organize value creation activities scattered in different regions [33]. Gereffi et al. believe that the governance of GVC should be studied based on three theories: transaction cost, enterprise network, and enterprise learning ability [22]. According to the complexity of market transactions and taking trading capacity and supply capacity as the standard, they subdivided the governance mode of GVC into

five types, namely market value chains, modular value chains, relational value chains, captive value chains, and hierarchy value chains [34]. Among the five governance models, market value chains and hierarchy value chains are at the lowest and highest end of the value chain behavior, respectively.

Based on the existing literature on the factors affecting the promotion of GVCs, this article summarizes the factors affecting the promotion of GVCs, mainly including production factor endowment, foreign direct investment, technological innovation, and policy factors. Lin and Chang believe that the number of various factors of production (including land, labor, capital, and entrepreneurial talent) owned by a country and available for production is the key factor in promoting regional GVsC [35]. Blyde's study found that FDI can promote the position of manufacturing GVCs, but the promotion effect of FDI is different in different positions of GVC [36]. Cario believes that technological innovation is an important factor in promoting the manufacturing industry's GVC [37]. Herzer believes that competition, market demand, technological progress, and policy factors are the external factors affecting the promotion of the GVCs [38].

### 2.3. The Impact of Digitalization on the Division of Labor and Its Status in the GVC of Manufacturing

Industrial digitization is based on digital technology. With information data as the key factor in production and data authorization as the main body, it realizes the digital upgrade of the industry [39]. Artificial intelligence, big data, and cloud computing based on digital technology have promoted traditional industries' intelligence, informatization, and technological nation and can effectively promote the transformation and collaborative innovation of traditional industry kinetic energy [40].

#### 2.3.1. The Impact Mechanism of Digital Technology on the Division and Status in the GVC

Research by Foster et al. focuses on how digital technologies affect the distribution of added value in the GVCs. Digital technology promotes the standardization of goods and services and improves the flexibility of GVCs. The added value obtained by participating in the division of labor in GVCs is increasingly related to digital technology [41]. Restructuring the GVC with digital technology will also affect the distribution of added value in each value chain link. Digitalization reduces the cost of information search and realizes product diversification. The added value of R&D, design, and sales also changes due to the investment in digital technology producing a "long tail effect" [42]. Digital technology accelerates the automation of production processes to obtain more added value. The use of robots impacts the added value of pre-production stages, such as software design and data services, and post-production stages, such as after-sales services, enabling intelligent enterprises to obtain more growth value [40]. The digital economy has become an important engine for manufacturing transformation and value chain upgrading.

Perez and Soete [43] clearly stated that "every technological revolution forms a suitable technological and economic paradigm". At present, it is in the changing period of iterative new technology and economic paradigm. The GVC concept proposed by Porter [18] provides a brand-new dimension for evaluating an economy's industrial advantages and enterprise competitiveness. This definition covers the core modules of power system, governance structure, and industrial upgrading. Kogut proposed that allocating each link of the whole value chain between different countries and regions depends on the advantages of different countries and regions. Countries choose the value chain segments according to their respective advantages [44]. Later, Gereffi et al. [22] used the 3C model to observe the dynamic changes in industrial structure. Kaplinsky [45] proposed "inter-industry value chain and intra-industry value chain" and "chain horizontal governance" industrial cluster from the perspective of endogenous complementarity, which continuously broadened the research boundary. More consensus is reached, and that practice has proved that developing countries are locked in low value-added links with low barriers in the

GVCs. The fierce competition causes the dilemma of "poverty growth", which will not lead to technological progress [24].

### 2.3.2. GVC Measurement Methods

Hummels et al. proposed to use the vertical specialization trade value (VS) and a country's export of goods as an input for the re-export of imported goods (VS1) as a measure [46]. Koopman et al. use VS and VS1 to measure the degree of GVC participation: taking the United States, China, Japan, Germany, and Russia as examples, the results of VS and VS1 ratios show that Germany's forward participation index is higher than other countries, and Russia's backward participation index is higher than that of other countries [9]. Based on the decomposition method of trade value added, this article studies the status and evolution of the international division of labor in the GVC from three levels: country, region, and industry [47]. The decomposition method of trade added value (WWYZ) is used to analyze the status and influencing factors of China's service industry, equipment manufacturing industry, and automobile industry in the division of labor in the GVC from the changes in vertical specialization rate, foreign added value ratio, and added value export ratio [10]. The share of vertical specialization (VSS) measures the international division of labor status. It is measured by the proportion of imported intermediate goods in a country's exports to total output. The indicator is that the larger the country's participation in the international division of labor is, the deeper it is [46]. The export technical complexity (EXPY) reflects the technical level of a country's specific industry. Taking China's equipment manufacturing industry as an empirical analysis, the larger the export technical complexity index value, the higher the technical level of the export products, and the higher the international division of labor [48].

### 2.3.3. Quantitative Measurement of the Impact of Digitalization on the Division of Labor in Manufacturing GVCs

As a measurement of industrial integration, manufacturing digitization has not yet formed a unified standard at home and abroad. The current academic methods for the measurement of manufacturing digitization include:

Industrial convergence measures the level of digitization. Zudaire E proposed the early digital measurement in the book "Production and Distribution of Knowledge in the United States" [49]. It determined the level of digitalization by measuring the proportion of the output value of the information industry in GDP. From the perspective of industrial integration, the proportion of the ICT industry input in the manufacturing industry to the total output is taken as an indicator of the level of digitalization [50].

The input–output method measures the level of digitization. According to the input–output relationship between departments in the input–output table, the digitalization level of the manufacturing industry is determined. The service level of the manufacturing industry is measured by the proportion of the investment in the service industry in the total investment in the manufacturing process [51].

The advantage of using the industrial integration method to calculate the digital level is that this method can more comprehensively summarize the current situation of the industrial digital level. The disadvantage is that the measured results are much larger than the actual situation. Because some non-digital industries cannot be distinguished, they are also included in the scope of digital industries. This method is suitable for analyzing the current situation of digitization. The advantage of using the input–output method to measure the digital level is that it can accurately reflect the current situation of the industrial digital level. The disadvantage is that the measured results are less than the actual situation because some digital industries are not included in the scope of measurement. This method is suitable for the detailed study of digitization [52].

2.3.4. Research on the Path and Strategy of Digital-Driven Manufacturing Value Chain Upgrade

Through innovation, digitization leads to the reduction in the transaction cost of the industrial chain due to the internal transformation into the external. It promotes the upgrading of the industrial GVC [22]. He, W. B et al. proposed strengthening the digital economy's participation in the GVC, broadening market channels, reducing transaction costs, increasing employment opportunities, and tapping economies of scale and scope. It helps realize value appreciation and makes economic entities more advantageous due to the integration of informatization and industrialization [53].

Through the above literature review, it can be seen that there are relatively few studies on the digitization of the manufacturing industry. The existing research results mostly discuss the impact of the digital economy on the national economy from a macroscopic perspective. The impact of the digital economy on the manufacturing industry is relatively small. Moreover, the quantitative indicators of manufacturing digitization have not formed a system in the academic community, so the research on manufacturing digitization needs to be improved. The relevant research on the GVCs has formed a complete system. Many scholars have measured China's division of labor position in the GVCs from the national, industrial, and other levels. They also discuss the factors affecting the division of labor status of China's GVCs. However, there are few studies on the influencing factors of China's GVCs from the digital economy perspective. The research on the impact of the digital economy on the division of labor position of manufacturing GVCs is in its infancy. Therefore, the research of this article has more theoretical significance and practical expansion space.

## 3. Digital Model Description and Numerical Analysis

### 3.1. Manufacturing Digital Model Establishment

This article uses the input–output method to measure the degree of digitization of manufacturing by the proportion of the added value of the ICT industry in the added value of manufacturing exports. However, given the differences in the statistical caliber and the availability of data in compiling input–output tables, this article focuses on the impact of digitalization on the division of labor in manufacturing GVCs. The specific model is [54]:

$$\text{Digitization of Manufacturing Index (DMI)} = \frac{\text{the added value of ICT industry}}{\text{the added value of manufacturing export}} \quad (1)$$

3.1.1. Measurement of the Digital Level in the Manufacturing Industry of Major Countries

The raw data comes from the World Input-Output Database (WIOD), widely used when calculating GVC-related indicators. The latest 2017 edition of the database covers 56 industry datasets from 43 economies worldwide, with consistent statistical standards and reliable data sources.

The added value of manufacturing exports and the ICT industry's added value are derived from the TIVA database constructed by the Organization for Economic Cooperation and Development (OECD)—World Trade Organization (WTO). The added value of manufacturing exports and export added value from 2005 to 2017 are derived from the ADB-PRIO database. For the convenience of analysis, 18 representative countries were selected from the 43 countries included in the WIOD database for comparative analysis, including developed countries (the United States, Japan, Germany, South Korea, the United Kingdom, France, Finland, Canada, Denmark, Norway, Sweden, Netherlands, etc.), and developing countries (China, Brazil, Russia, India, Mexico, Indonesia). According to the changes in recent years, South Korea's manufacturing industry has the highest degree of digitization, China's manufacturing industry has developed rapidly, Mexico and Japan have a higher degree of digitization, and the United States, the United Kingdom, Germany, France, and Sweden have a higher degree of digitization in the overall manufacturing industry, with stable development. The digitalization of the service industry is high, which

is mainly related to the advantages of traditional manufacturing in developed countries [54] (Table 1).

**Table 1.** Comparative analysis of the digitalization level of manufacturing in major countries (%).

| Country / Year | 2005 | 2007 | 2009 | 2011 | 2013 | 2015 | 2017 |
|---|---|---|---|---|---|---|---|
| Brazil | 5.26 | 4.25 | 4.03 | 3.56 | 3.6 | 3.58 | 3.61 |
| Canada | 4.96 | 4.73 | 5.16 | 4.03 | 4.22 | 4.48 | 4.62 |
| China | 19.07 | 17.51 | 16.81 | 14.81 | 15.69 | 16.55 | 17.57 |
| Germany | 9.26 | 9.25 | 10.91 | 9.82 | 0.01 | 9.66 | 9.56 |
| Denmark | 8.62 | 8.77 | 8.26 | 7.83 | 7.29 | 7.49 | 7.82 |
| Finland | 22.98 | 20.53 | 19.44 | 12.9 | 11.17 | 12.17 | 13.01 |
| France | 9.97 | 9.22 | 9.11 | 8.01 | 8 | 7.9 | 8.05 |
| UK | 9.62 | 8.6 | 8.92 | 8.07 | 7.8 | 8.37 | 8.35 |
| USA | 13.24 | 11.48 | 12.37 | 10.05 | 10.85 | 11.93 | 12.7 |
| Norway | 8.07 | 7.87 | 8.53 | 7.25 | 8.15 | 8.25 | 8.32 |
| Sweden | 12.89 | 11.22 | 12.55 | 9.89 | 10.07 | 7.93 | 7.96 |
| Netherlands | 7.5 7 | 7.29 | 7.47 | 6.58 | 6.56 | 6.89 | 7.01 |
| Indonesia | 6.67 | 5.39 | 6.06 | 4.79 | 4.82 | 5.2 | 5.12 |
| India | 2.97 | 3.1 | 3.73 | 2.61 | 2.6 | 3.57 | 3.55 |
| Japan | 16.92 | 15.17 | 14.34 | 12.53 | 12.08 | 12. | 12.6 |
| South Korea | 24.61 | 22.5 | 21.31 | 17.87 | 19.46 | 22.45 | 26.74 |
| Mexico | 17.5 | 16.34 | 17.91 | 14.07 | 14.82 | 16.68 | 16.72 |
| Russia | 2.87 | 2.75 | 3.54 | 2.76 | 2.66 | 3.31 | 3.40 |

Data source: Calculated based on the TIVA database and the National Bureau of Statistics.

3.1.2. Measurement of Digital Industry Level in China

In 2020, the total scale of China's digital economy reached CNY 39.2 trillion, accounting for 38.6% of GDP, with a growth rate of 8.2%, ranking second in the world [3]. From the perspective of digital industrialization infrastructure, it mainly includes 5G, integrated circuits, artificial intelligence, big data, cloud computing, blockchain, and other electronic information industries and the Internet, providing technical products and services for the digital economy. Among the three industries, the digitalization degree of the service industry is the highest at 37.8%, the digital industrial economy is in the middle at 19.5%, and the digitalization of the agricultural industry is the lowest. From 2017 to 2020, the proportion of industrial digitalization in GDP continued to rise, accounting for 17.22%, 18.31%, 19.5%, and 21% of the industry's added value. The trend of digitalization accelerating integration and penetration is obvious. However, the degree of digitalization within each industrial department shows structural differentiation. In 2018, China's ICT manufacturing industry increased by 13.7%. The top 20 sectors with digitalization degrees are shown in Figure 1, and the power transmission, distribution, and control equipment sector is the largest at 24.2%. The instant food sector is the lowest at 5%.

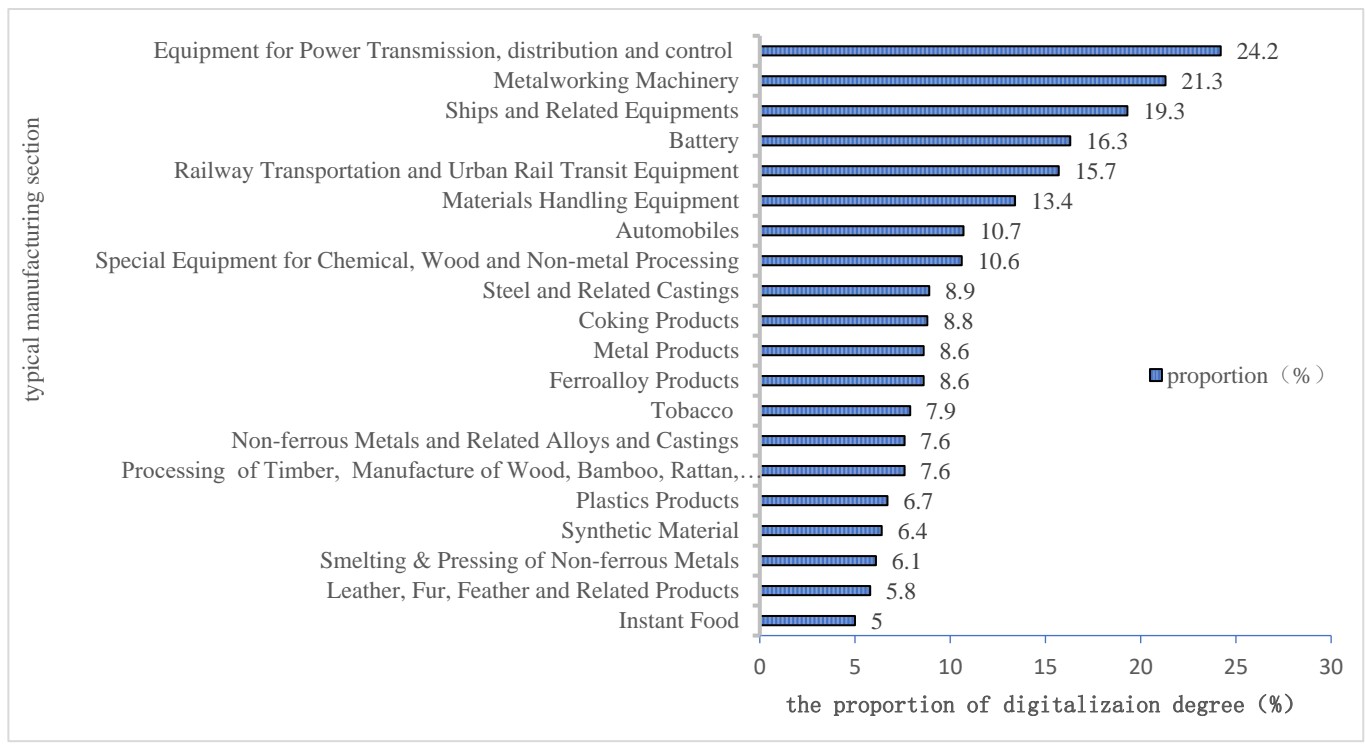

**Figure 1.** Proportion of digitalization of typical sectors of China's manufacturing industry in 2018. Data source: Calculated based on the TIVA database and the National Bureau of Statistics.

### 3.1.3. Measurement of the Industrial Digitalization Level in China

Data is a key element of the new generation of digital technology. Industrial Internet, intelligent manufacturing, Internet of vehicles, and other integrated new industries are the main content of the new generation of digital technology. At the same time, value release is its core and it is enabling. Data integration, platform empowerment, and other new-generation technologies lead the upgrading and transformation of the entire industrial chain. According to data from the China Academy of Information and Communications Technology, China's industrial digitalization's added value reached CNY 31.8 trillion in 2020, accounting for 31.2% of GDP. The fields of new infrastructure construction represented by industrial Internet, intelligent transportation, and 5G are growing rapidly [52].

As shown in Figure 2, according to The Global Industry Research Institute of Tsinghua University's "Research Report on the Digital Transformation of Chinese Enterprises (2020)" [12], due to the impact of COVID-19, the growth rate of digital transformation of the manufacturing industry is under increasing pressure, with a year-on-year increase of 9.3%. The market size has reached CNY 245.5 billion [54]. Completing the intelligent and service-oriented transformation of products, i.e., intelligent service, is the focus of the digital transformation of the machinery and equipment industries. The cloud computing platform has spawned new models and a new form of industry. The industrial internet platform has become a powerful tool for the manufacturing industry's digital transformation. (Companies such as Alibaba, Huawei, and Siemens are successful cases.) The industrial internet platform creates an open and shared value network. The key breakthrough in platform development is industrial platform-as-a-service (PaaS).

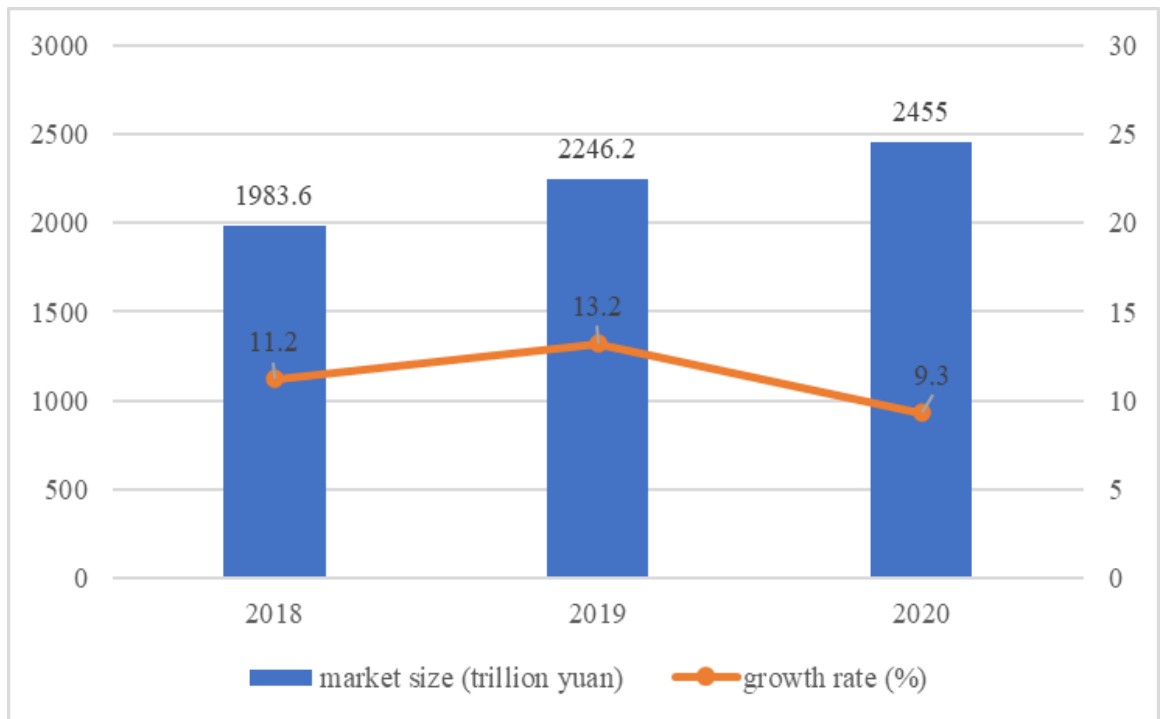

**Figure 2.** Market scale and growth rate of China's digitization of manufacturing in 2018–2020. Data source: Calculated based on the TIVA database and the National Bureau of Statistics.

*3.2. Comparative Analysis of the Digital Advantages of Manufacturing in China and Major Countries in the World*

In 2020, the total scale of global digital economy development reached 30.2 trillion US dollars, accounting for 40.3% of global GDP [12]. The United States ranks first with USD 12.34 trillion, and China has USD 4.73 trillion [55]. As shown in Figure 3, from the scale of digital industrialization, the United States is USD 1.5236 trillion, followed by China with USD 968.9 billion. The digital transformation of traditional manufacturing has become the dominant trend. Among them, Germany is the highest, reaching more than 90%. More than 10 countries, including the United Kingdom, the United States, and Russia, also accounted for more than 80%, and China accounted for 79.31%. From the perspective of the level of industrial digitalization, as shown in Figure 4, in 2019, this index and the industry added value accounted for more than 1/3 in South Korea, Germany, the United States, the United Kingdom, and Japan. Among them, South Korea is the highest at 45% [56]. China's industrial digitalization accounts for only 18.3% of the industry's added value, clearly in the initial stage of digital transformation. In terms of the quality and level of digitalization, the focus of major national applications is to promote the effectiveness of production process management and control based on 5G and the industrial Internet of things platform as well as to tap the potential of data, indicating that the quality and depth of digitalization are high [54]. However, China is limited by the unbalanced development of digitalization, many small, medium, and micro-enterprises, and the weak foundation of the industry, so some applications remain in the links of visual description monitoring and diagnosis (Figure 4).

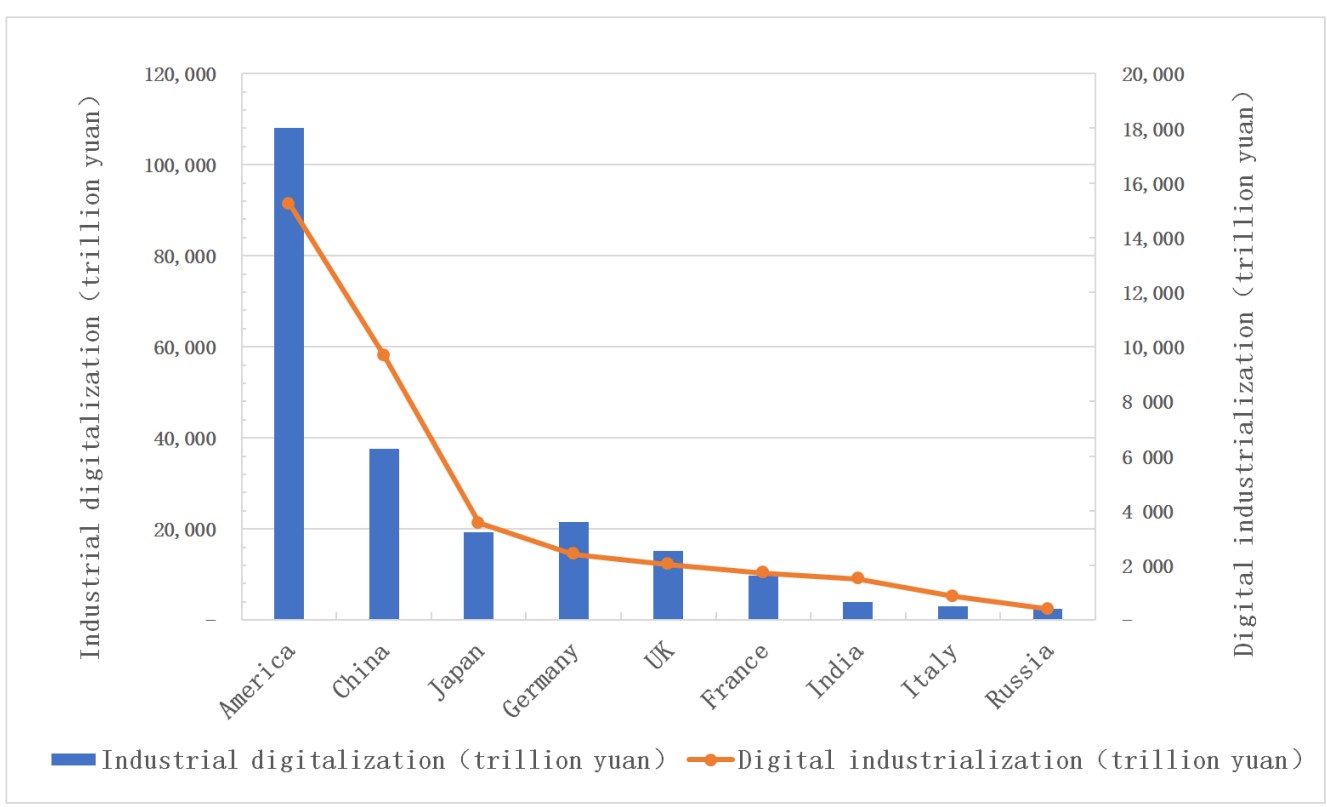

**Figure 3.** The scale of digital industrialization and industrial digitalization in various countries. Data source: Calculated based on the TIVA database and the National Bureau of Statistics.

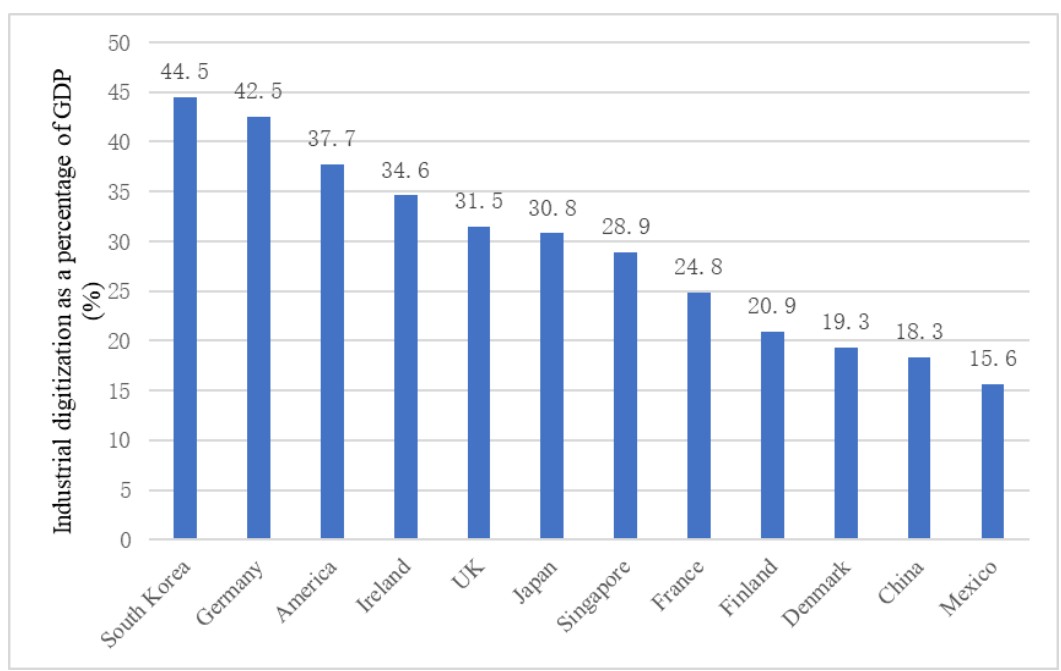

**Figure 4.** Industrial digitization as a percentage of GDP. Data source: Calculated based on the TIVA database and the National Bureau of Statistics.

## 4. Manufacturing Global Value Chain Participation Model and Data Analysis

### 4.1. The Structure Equation and Variables

China is increasingly integrated into the GVC system in the international division of labor. Existing literature studies track a lot of China's participation in the GVCs' division

of labor, mainly focusing on the participation and its position in the GVCs. The relevant research uses data from different industries to measure China's degree of participation in the GVCs' division of labor [57]. The results show that most of China's manufacturing industries are engaged in the assembly of imported parts and components located in the relative downstream division of GVCs. Overall, China's position in the GVC division is constantly improving [58].

It is generally accepted that Koopman et al. used the Koopman, Powers, Wang, and Wei (KPWW) method to decompose the WIOD input–output table [9]. In 2013, the OECD extended three indicators of GVC participation, length, and distance to final demand. This article adopts the 2018 ADB-MRIO database and the WWYZ decomposition method of trade added value based on the KPWW method proposed by Wang et al. [59]:

$$
\begin{aligned}
(\mathrm{Va^s})' = \; & \widehat{V}^s L^{SS} Y^{SS} + \widehat{V}^s L^{SS} E^{S*} + \widehat{V}^s L^{SS} Y^{SS} + \widehat{V}^s L^{SS} \textstyle\sum_{r \neq s}^M Y^{sr} + \widehat{V}^s L^{SS} \sum_{r \neq s} A^{Sr} \sum_U^M (B^{ru} \sum_t^M Y^{ut}) = \mathrm{DVA_{INT}} + \\
& \mathrm{DVA_{INTrex}} + (\mathrm{RDV}) + (\mathrm{FVA_{FIN}} + \mathrm{FVA_{INT}}) + (\mathrm{DDC} + \mathrm{FDC}) = \mathrm{DVA} + \mathrm{RDV} + \mathrm{FVA} + \mathrm{PDC}
\end{aligned}
\tag{2}
$$

The total export added value of the manufacturing industry is composed of DVA, RDV, FVA, and PDC, where A and B are the direct consumption coefficient matrix and complete consumption coefficient matrix, respectively. Y and X are the final demand and the total output matrix, respectively. L is the domestic Leontief inverse matrix, and V represents the directly added value coefficient matrix. The above formula shows that, at the sectoral level, the total exports of country *s* to country *r* can be completely decomposed into 16 terms in 8 categories and can finally be combined into four categories of variables:

(1) DVA: This represents the domestic added value ultimately absorbed abroad. When decomposed, DVA_FIN represents the domestic added value exported in the form of final products; DVA_INT represents the domestic added value exported in the form of intermediate products and produced by the importing country for final demand; DVA_INTrex represents the domestic added value that is processed and produced by the importing country in the form of intermediate products and exported to the third country and finally consumed.

(2) RDV: This represents the country's domestic added value exported and then returned and consumed.

(3) FVA: This represents foreign added value in domestic exports: where FVA_FIN represents foreign added value included in final product exports, and FVA_INT represents foreign added value included in intermediate product exports.

(4) Pure double counting (PDC): This is caused by the trade of intermediate products crossing national borders. DDC stands for double counting from domestic accounts, and FDC stands for double counting from foreign accounts. Based on the decomposition framework of trade added value, Koopman et al. constructed a GVC status index and a participation index to measure a country's GVC division of labor and its participation in a specific sector. The GVC status index is given as follows [9]:

$$
\mathrm{GVC\ participation}_{is} = \ln\left(1 + \frac{\mathrm{IV_{is}}}{\mathrm{E_{is}}}\right) - \ln\left(1 + \frac{\mathrm{FV_{is}}}{\mathrm{E_{is}}}\right)
\tag{3}
$$

where GVC participation$_{is}$ indicates the position index of the i industry in country s in GVC. IV$_{is}$ indicates the indirect export added value of the i industry in country s, that is, the domestic added value in the form of intermediate products processed and produced by the importing country and exported to a third country and finally consumed. FV$_{is}$ represents the foreign added value included in the export of i industry in country s. E$_{is}$ represents the total export value of the industry in country *s*. In the GVC division of labor, if the proportion of IV in exports of a country is higher than that of FV in exports, the country mainly participates in the international division of labor by exporting intermediate products or services to other countries, indicating that the country is in the upstream link of GVC. On the contrary, the proportion of

IV in exports of a country is lower than that of FV in exports. The country mainly participates in the international division of labor by importing intermediate products from other countries and then processing, assembling, and re-exporting, indicating that the country is in the downstream link of GVC.

According to Equation (3), the key to calculating the GVC position index is the proportion of IV and FV in exports, so multiple countries with differences in IV, FV, and E may also have the same GVC position index. For this reason, Koopman et al. also constructed a GVC engagement index with the following equation [10]:

$$\text{GVC participation}_{is} = \frac{IV_{is}}{E_{is}} + \frac{FV_{is}}{E_{is}} \tag{4}$$

where $\frac{IV_{is}}{E_{is}}$ is the proportion of the domestic indirect added value of the i sector in country s to the total export of the i sector, indicating the forward participation of the sector in the international division of labor, $\frac{FV_{is}}{E_{is}}$ is the proportion of the foreign added value of the i sector in the country s to the total export of the ith sector, which indicates the backward participation of the sector in the international division of labor. A high degree of forwarding participation indicates that a country mainly participates in the international division of labor by exporting intermediate products and services to other countries. Furthermore, a high degree of backward participation indicates that a country mainly participates in the international division of labor by importing intermediate products and services and faces a value chain "low-end lock-in" threat. The first item is the domestic added value of domestic production and consumption that is not involved in the division of labor in international trade. The second item is included in final product exports. Therefore, the forward decomposition is the proportion of the added value of international trade in a certain sector to the whole sector. The backward decomposition is the proportion of the added value of the final product in the entire sector. The sum of the ratios is the participation in the GVCs, reflecting the reality of the refinement of the global professional division of labor.

*4.2. Data Analysis*

According to the data in the world input–output database, in 2000, it was USD 16.018 billion, it increased to USD 157.577 billion in 2014, and finally it increased to USD 371.231 billion in 2017, achieving a compound annual growth rate of 20.31%, consistent with the growth of total export scale [60]. Its proportion shows a "U"-shaped change. It shows that the transformation and upgrading of the manufacturing industry are accelerating. Moreover, China's manufacturing RDV ratio increased from USD 163 million in 2000 to USD 7.93 billion in 2017, accounting for about 1%, which reflects the added value of products that are first exported abroad and then returned to be consumed in China for the final consumption being relatively low. In 2000, the scale of China's FVA was USD 3.084 billion, accounting for about 15%. In 2017, it reached USD 318.257 billion, an increase of about 40%, indicating that China's manufacturing industry has increased its dependence on foreign countries in the process of digital transformation [60].

4.2.1. Increase in PDC Value

In 2000, the PDC value was USD 16.672 billion, and it increased to USD 66.652 billion in 2014. It decreased to USD 32.429 billion in 2017, reflecting the fact that the development of China's manufacturing industry is struggling under the background of superimposed impacts, such as increased global production and operation costs, economic recession, and increased trade protection (Figure 5a) [61].

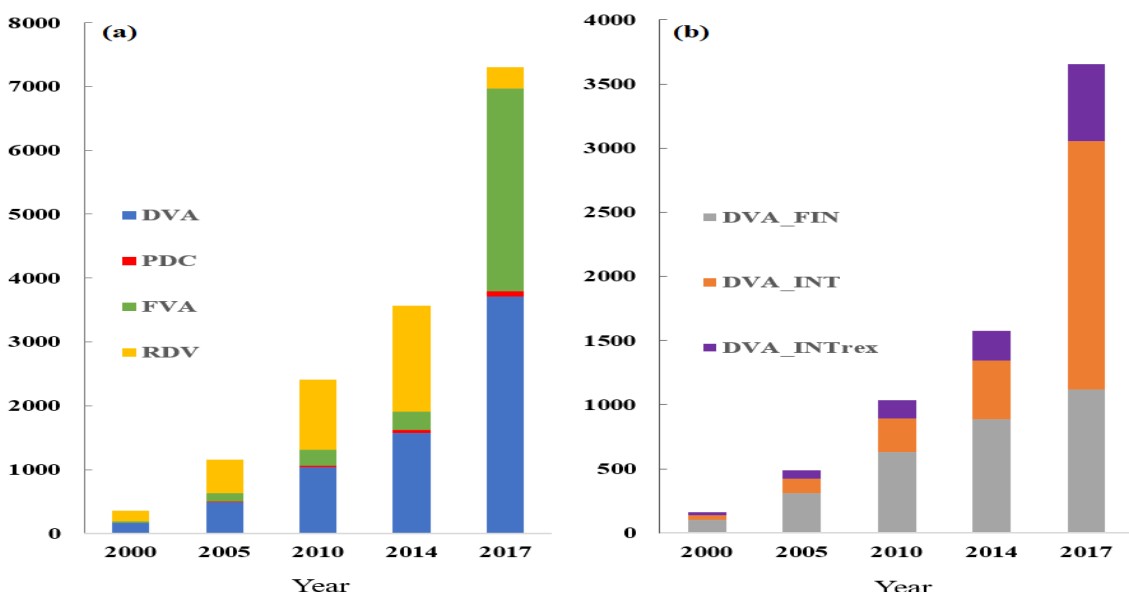

**Figure 5.** The decomposition of (**a**) added value of total exports and (**b**) domestic trade in billions of US dollars. Data source: ADB-MRIO2018 and UIBE GVC indicators. Data were acquired from the TIVA database and the National Bureau of Statistics.

In addition, in the decomposition of the domestic added value of China's manufacturing industry, as shown in Figure 5b, DVA_FIN, the added value of final export and direct consumption contributes the most. Still, its proportion decreased from 50% in 2000 to 45% in 2014 and 31.71% in 2017. DVA_INT, the added value of export to other countries as intermediate goods, increased from 18% in 2000 to 23% in 2014 and 52.19% in 2017. The added value DVA_INTREX, exported as intermediate goods to other countries and re-exported, increased from 11% in 2000 to 12% in 2014 and 16.1% in 2017, reflecting that China's manufacturing exports are engaged in processing. The assembly of imported intermediate goods and other low value-added conditions improved slightly. The above results show that China's manufacturing exports in processing and assembling imported intermediate goods and other low added value status improved slightly (Figure 5b) [62].

4.2.2. Analysis of GVC Status Index in Manufacturing Countries

According to the calculation results of Equations (3) and (4), it can be seen from Table 2 that in terms of time series changes, the manufacturing GVC status index of France, the United Kingdom, Canada, Sweden, Estonia, and Mexico maintained a continuous downward trend from 2010 to 2017. This indicates that the manufacturing industry's international division of labor status in these countries continued to decline. As export-oriented countries, these countries have a relatively high degree of economic dependence on foreign countries and are more vulnerable to the current global economic downturn.

**Table 2.** Comparison of manufacturing GVC status and participation index of major countries.

| Country / Year | Status Index | | | | | Participation Index | | | | |
|---|---|---|---|---|---|---|---|---|---|---|
| | 2010 | 2014 | 2015 | 2016 | 2017 | 2010 | 2014 | 2015 | 2016 | 2017 |
| China | −0.048 | −0.009 | −0.007 | 0.011 | 0.018 | 0.282 | 0.260 | 0.245 | 0.246 | 0.247 |
| Canada | −0.158 | −0.172 | −0.189 | −0.188 | −0.184 | 0.369 | 0.383 | 0.393 | 0.392 | 0.391 |
| Brazil | 0.064 | 0.040 | 0.012 | 0.027 | 0.039 | 0.283 | 0.307 | 0.305 | 0.292 | 0.292 |
| Germany | −0.058 | −0.060 | −0.064 | −0.046 | 0.055 | 0.364 | 0.368 | 0.374 | 0.365 | 0.370 |
| France | −0.084 | −0.087 | −0.084 | −0.102 | −0.102 | 0.399 | 0.405 | 0.398 | 0.415 | 0.415 |
| UK | −0.060 | −0.038 | −0.045 | −0.054 | −0.058 | 0.393 | 0.383 | 0.387 | 0.391 | 0.397 |
| Denmark | −0.116 | −0.130 | −0.136 | −0.121 | −0.121 | 0.394 | 0.413 | 0.419 | 0.407 | 0.407 |
| USA | 0.035 | 0.018 | 0.030 | 0.036 | 0.031 | 0.252 | 0.263 | 0.251 | 0.245 | 0.249 |
| Japan | 0.063 | 0.007 | 0.002 | 0.030 | 0.017 | 0.321 | 0.353 | 0.342 | 0.329 | 0.339 |
| South Korea | −0.115 | −0.108 | −0.125 | −0.106 | −0.108 | 0.420 | 0.421 | 0.428 | 0.419 | 0.42 |
| Estonia | −0.169 | −0.201 | −0.204 | −0.214 | −0.214 | 0.480 | 0.500 | 0.504 | 0.507 | 0.507 |
| India | −0.041 | −0.045 | −0.026 | 0.023 | −0.019 | 0.372 | 0.362 | 0.337 | 0.316 | 0.341 |
| Indonesia | 0.041 | 0.018 | 0.028 | 0.056 | 0.041 | 0.334 | 0.343 | 0.323 | 0.297 | 0.323 |
| Mexico | −0.266 | −0.245 | −0.240 | −0.292 | −0.276 | 0.447 | 0.427 | 0.412 | 0.464 | 0.45 |
| Norway | 0.035 | 0.045 | 0.021 | 0.024 | 0.039 | 0.400 | 0.406 | 0.403 | 0.399 | 0.405 |
| Russia | 0.223 | 0.187 | 0.179 | 0.204 | 0.202 | 0.374 | 0.379 | 0.383 | 0.386 | 0.385 |
| Sweden | −0.062 | −0.051 | −0.055 | −0.066 | −0.070 | 0.420 | 0.416 | 0.420 | 0.429 | 0.427 |

Data source: Calculated based on data from the ADB database.

The division of labor in the GVCs of Germany, Denmark, South Korea, Japan, and the United States has improved, and the GVC status index showed an upward trend in fluctuations. The main reasons are that these countries have issued support policies in the process of returning to high-end manufacturing and vigorously developing high-tech industries. On the other hand, these countries implement trade protectionism, thus encouraging exports and restricting imports. This results in a relative increase in forwarding participation and a gradual decrease in backward participation. The manufacturing GVC status index of China, Indonesia, Norway, and Russia continued to rise, indicating that these countries have shown a positive upward trend toward the mid-to-high end of GVC in recent years. In terms of participation index, manufacturing GVCs in Brazil, Denmark, Japan, and the United States declined from 2014 to 2017, mainly because the GVC participation index in these countries increased in the forward participation and the decline of the backward embedding degree is greater than the increase in forwarding embedding degree. From a vertical perspective, the manufacturing GVC status index of Brazil, the United States, China, and Japan is still at the forefront of the sample countries. In 2015, China's manufacturing GVCs' division of labor status index changed from negative to positive and reached 0.018 in 2017, surpassing Japan. The GVC status index of export-oriented economies, such as Canada, Denmark, and South Korea, lags far behind other countries. The GVC participation index of these countries is at the forefront of the sample countries, mainly because they participate in the international division of labor in the way of backward participation, and they are in a lower position in the GVCs. The manufacturing GVC status index and participation index of the remaining countries are at the middle level of the sample countries [63].

*4.3. Analysis of GVC Participation Index in the Manufacturing Sector*

4.3.1. Model Establishment

The equations used are as follows:

$$GVC_{part} - FOR_{d,t} = \frac{V\_GVC\_R^s}{\tilde{A}^{TM}V^dX^d} + \frac{V\_GVC\_D^s}{V^d\tilde{A}^{TM}X^d} + \frac{V\_GVC\_F^s}{V^d\tilde{A}^{TM}X^d} \tag{5}$$

$$GVC_{part} - BACK_{s,t} = \frac{V\_GVC\_R^s}{Y^s} + \frac{Y\_GVC\_D^s}{Y^s} + \frac{V\_GVC\_F^s}{Y^s} \tag{6}$$

where $\frac{V\_GVC\_R^s}{\tilde{A}^{TM}V^dX^d}$ represents the domestic added value of domestic production and consumption that does not participate in the division of labour in international trade; $\frac{V\_GVC\_D^s}{V^d\tilde{A}^{TM}X^d}$ is domestic added value included in final product exports. V_GVC_D indicates that the exporting country exports intermediate products first, and the implied added value returns to the exporting country. V_GVC_F represents the domestic added value of an exporting country re-exported to a third country as an intermediate product [60].

According to the various intensity of different manufacturing factors, we divide the industry into three groups: labor, capital, and technology-intensive groups; then, we use Equations (5) and (6) to analyze the participation index of the GVC (Table 3).

**Table 3.** WIOD manufacturing classification comparison table.

| Code | Name | Element Density Classification |
|---|---|---|
| R5 | Food, Beverage, and Tobacco Manufacturing | Labor intensive |
| R6 | Textiles, Apparel, and Leather Goods | |
| R7 | Manufacture of woven material products | |
| R8 | Paper and Paper Products | |
| R9 | Printing and copying of recording media | |
| R22 | Furniture Manufacturing; Other Manufacturing | |
| R10 | Coke and Refined Petroleum Products | Capital intensive |
| R11 | Chemicals and chemical products | |
| R12 | Essential pharmaceutical products and pharmaceutical preparations | |
| R13 | Rubber and Plastic Products | |
| R14 | Other non-metallic minerals | |
| R15 | Manufacture of base metals | |
| R16 | Manufacture of metal products, other than machinery and equipment | |
| R17 | Manufacture of computer, electronic, and optical products | Technology intensive |
| R18 | Electrical equipment manufacturing | |
| R19 | Machinery and equipment manufacturing | |
| R20 | Automobile, Trailer, and Semi-trailer Manufacturing | |
| R21 | Manufacturing of other transportation equipment | |

Data source: Calculated based on data from the ADB database.

4.3.2. Data Analysis

According to Equations (5) and (6), the forward decomposition of the production decomposition of the added value to the industry GDP and the backward decomposition of the final product production decomposition is obtained. The ratio of the two is the participation in the GVCs. From 2000 to 2017, the GVC participation of various sectors of China's manufacturing sector is between 0.041 and 3.56, which is basically in the middle and lower part of the "smile curve" of the manufacturing industry chain. Except for D15, D17, and D18, D05, D06, D07, D08, D13, and D14 all showed a decline from 2000 to 2008, and then achieved a U-shaped rise in 2017, whereas the other departments all increased steadily (Table 4). The results show that the status of China's manufacturing value chain is continuously and slowly improving.

**Table 4.** GVC participation index of China's manufacturing sector.

| Year | 2000 | | | 2008 | | | 2014 | | | 2017 | | |
|---|---|---|---|---|---|---|---|---|---|---|---|---|
| Sector | Participation | Easy | Complex | Participation | Easy | Complex | Participation | Easy | Complex | Participation | Easy | Complex |
| Food and beverage manufacturing and tobacco industry d05 | 0.49 | 0.12 | 0.05 | 0.43 | 0.96 | 0.22 | 0.72 | 1.21 | 0.42 | 0.77 | 1.11 | 0.50 |
| Textile, apparel, and leather product manufacturing d06 | 0.73 | 0.12 | 0.22 | 1.09 | 1.55 | 0.67 | 1.08 | 1.25 | 0.84 | 1.11 | 1.16 | 1.05 |
| Wood, wood products processing industry, and bamboo, yard cloth products d07 | 0.97 | 0.19 | 0.14 | 0.90 | 1.05 | 0.74 | 1.13 | 1.59 | 0.76 | 1.15 | 1.31 | 0.95 |
| Paper and paper products industry d08 | 0.87 | 0.24 | 0.13 | 0.74 | 0.66 | 0.90 | 0.65 | 0.53 | 0.95 | 0.65 | 0.51 | 1.08 |
| Printing and reproduction of recording media dC09 | 0.85 | 0.22 | 0.11 | 1.05 | 1.26 | 0.84 | 1.07 | 1.16 | 0.97 | 1.09 | 1.08 | 1.11 |
| Coking and petroleum processing d10 | 0.81 | 0.26 | 0.19 | 1.14 | 2.94 | 0.60 | 1.33 | 4.45 | 0.65 | 1.37 | 2.96 | 0.78 |
| Chemical raw materials and chemical products manufacturing d11 | 1.01 | 0.28 | 0.22 | 0.54 | 1.24 | 0.23 | 0.64 | 1.41 | 0.27 | 0.66 | 1.50 | 0.28 |
| Basic pharmaceutical industry and pharmaceutical preparation industry d12 | 0.54 | 0.13 | 0.09 | 1.03 | 1.30 | 0.81 | 0.87 | 1.10 | 0.67 | 0.90 | 1.04 | 0.76 |
| Rubber and plastic products d13 | 1.06 | 0.23 | 0.30 | 0.70 | 0.88 | 0.55 | 0.79 | 0.88 | 0.69 | 0.84 | 0.90 | 0.77 |
| Non-metallic mineral products d14 | 0.61 | 0.15 | 0.14 | 0.73 | 1.82 | 0.46 | 0.98 | 2.22 | 0.60 | 1.01 | 1.88 | 0.68 |
| Basic metal products industry d15 | 0.98 | 0.25 | 0.18 | 0.58 | 0.60 | 0.54 | 0.50 | 0.49 | 0.52 | 0.52 | 0.50 | 0.58 |

**Table 4.** *Cont.*

| Year Sector | 2000 | | | 2008 | | | 2014 | | | 2017 | | |
|---|---|---|---|---|---|---|---|---|---|---|---|---|
| | Participation | Easy | Complex | Participation | Easy | Complex | Participation | Easy | Complex | Participation | Easy | Complex |
| Welded metal products industry d16 | 1.32 | 0.26 | 0.21 | 1.04 | 4.28 | 0.43 | 1.48 | 6.04 | 0.61 | 1.53 | 3.92 | 0.73 |
| Computers, electronic products, and optical product manufacturing d17 | 0.66 | 0.12 | 0.52 | 0.73 | 0.58 | 1.13 | 0.71 | 0.58 | 1.10 | 0.73 | 0.58 | 1.14 |
| Electronic equipment manufacturing d18 | 0.82 | 0.21 | 0.23 | 0.04 | 0.04 | 0.04 | 0.06 | 0.06 | 0.06 | 0.07 | 0.07 | 0.07 |
| Machinery and equipment manufacturing d19 | 0.56 | 0.19 | 0.14 | 0.70 | 0.88 | 0.55 | 0.79 | 0.88 | 0.69 | 0.84 | 0.90 | 0.77 |
| Automobile, trailer, and semi-trailer manufacturing d20 | 0.52 | 0.22 | 0.11 | 2.28 | 2.22 | 2.38 | 2.99 | 3.08 | 2.86 | 3.14 | 3.06 | 3.26 |
| Other transportation equipment manufacturing d21 | 0.55 | 0.20 | 0.16 | 2.31 | 2.28 | 2.36 | 3.16 | 3.24 | 3.04 | 3.31 | 3.23 | 3.56 |
| Other manufacturing d22 | 0.85 | 0.1 | 0.15 | 1.12 | 0.99 | 1.47 | 1.11 | 0.91 | 1.62 | 1.11 | 0.90 | 1.82 |

Data source: Calculated based on data from the ADB database.

It is shown in Table 5 that from 2000 to 2017, the forward participation of most industries in the manufacturing industry has increased, wheras the backward participation of most industries has declined. In participating in the GVC division of labor, China reduces its dependence on foreign added value and exports domestic added value. The network participation model of some mid-to-high-end manufacturing sectors has gradually shifted from "bottom embedded" to "high-level penetration". China already has a strong independent production capacity in the middle and high-end manufacturing sectors, such as electronics, petrochemicals, and machinery and equipment.

**Table 5.** China's manufacturing industry trade added value participation.

| Industry | Forward Participation Index | Backward Participation Index | Industry | Forward Participation Index | Backward Participation Index |
|---|---|---|---|---|---|
| | Δ (2017–2000) | | | Δ (2017–2000) | |
| Agriculture, forestry, animal husbandry, and fisheries | 0.009 | −0.008 | Water, electricity, and gas | 0.006 | −0.003 |
| Mining and quarry | −0.034 | 0.019 | Building industry | 0.001 | −0.015 |
| Food, drink, and tobacco | 0.002 | −0.007 | Wholesale trade | 0.002 | −0.053 |
| Textile and apparel | 0.017 | −0.057 | Retail trade | 0.007 | −0.053 |
| Leather and shoes | 0.023 | −0.161 | Accommodation and catering | −0.017 | −0.018 |
| Wood products industry | 0.02 | −0.021 | Inland transport | −0.005 | −0.005 |
| Paper printing | 0.002 | −0.019 | Surface transport | 0.031 | −0.017 |
| Coke oil | −0.032 | 0.052 | Airfreight | −0.041 | 0.035 |
| Chemicals | 0.009 | −0.024 | Paratransit | 0.051 | −0.028 |
| Rubber plastic | 0.004 | −0.041 | Post and telecommunications | −0.039 | −0.035 |
| Non-metallic products | 0.007 | −0.002 | Financial intermediary | 0.004 | −0.007 |
| Metallic products | −0.01 | 0.009 | Leasing industry | −0.042 | −0.051 |
| Machine made | 0.044 | −0.013 | National defense and social security | 0.006 | −0.024 |
| Electron optics | 0.037 | −0.048 | Education | 0 | −0.031 |
| Transportation equipment | −0.005 | −0.025 | Health and social work | −0.002 | −0.025 |
| Other manufacturing | 0.065 | −0.025 | Social service | −0.022 | −0.061 |

Data source: Calculated based on data from the ADB database.

The results in Table 5 show that the supporting service industries, such as transportation, finance, and trade, have also developed and expanded, showing a trend of increasing forward participation and decreasing backward participation.

### 4.4. Digitalization of the GVC of the Manufacturing Industry Fixed-effects Empirical Model

4.4.1. Model Construction

This article, considering the correlation and availability of data, draws on the related research to construct a theoretical model as follows:

$$\text{GVC}_{\text{pt\_f}_{d,t}} = \beta_0 + \beta_1 \text{digin} + \gamma C_{d,t} + v_d + \theta_t + \varepsilon_{d,t} \tag{7}$$

$$\text{GVC}_{\text{pt\_f}_{d,t}} = \beta_0 + \beta_1 \text{digin}_{d,t} + \beta_2 \text{digin}_{d,t} \times \text{numb}_{d,t} + \beta_3 \text{digin}_{d,t} \times \text{rede}_{d,t} + \gamma C_{d,t} + v_d + \theta_t + \varepsilon_{d,t} \tag{8}$$

where d and t represent department and year, respectively, $\beta_0$ is intercept term, $v_d$ is the individual fixed effect, $\theta_t$ is time fixed effect, and $\varepsilon_{d,t}$ is the residual term.

$\text{GVC}_{\text{pt\_f}_{d,t}}$ is GVC status index. The model controls industry fixed effects and time fixed effects.

### 4.4.2. The Hypothesis of the Expected Effects of the Variables

Referring to the existing research results on the influencing factors of the GVCs' promotion in the manufacturing industry and combined with the GVCs' promotion theory in the manufacturing industry, abro, scal, capi, intu, prod, numb, and RD are set as control variables in this article. Firstly, numb, capi and scal indicate the production factor endowment of China's manufacturing industry. Many research results show that the endowment of high-quality production factors can effectively reduce the production cost of the manufacturing industry, which is the basis for improving GVC of the manufacturing industry. Secondly, prod, RD, and intu indicate China's manufacturing industry's technological development level and scientific and technological innovation investment. Technology leadership and innovation can not only reduce production costs but also improve product quality and reduce external dependence. It is the key to improving GVC in the manufacturing industry. Thirdly, abro shows that an increase or decrease in the market demand of the manufacturing industry will directly affect the production of manufacturing enterprises. It is the endogenous driving force for improving GVC in the manufacturing industry.

We have introduced digital input (Digin), overseas demand (Abro), output scale (Scal), nominal capital stock (Capi), added value divided by the size of employed human capital (Prod), number of manufacturing enterprises (numb), and R&D spending divided by main business in-come (Rd) to validate the regression results and hypothesize that these variables have a positive effect on improving the level of GVCs. The meaning of each variable is listed in Table 6.

**Table 6.** Basis for variable selection and data sources; interpretation of expected effects.

| | Variables | Explanation | Data Source | Expected Sign |
|---|---|---|---|---|
| Be explained variable | $GVC_{pt\_f_{d,t}}$ | Forward contact engagement degree | ADB-MRIO2018 edition collation | |
| Explanatory variables | Digin | Digital input: It is measured by the added value of communications and information services, the latter including software services, circuit design, and testing services, information system services, and business process management services. | ADB-MRIO2018 | + |
| Control variables | Abro | Overseas demand: sectoral added value | ADB-MRIO2018 | + |
| | Scal | the sectoral added value output | ADB-MRIO2018 | + |
| | Capi | nominal capital stock | | + |
| | Intu | Return on capital divided by return on labor | | + |
| | Prod | added value divided by the size of employed human capital | China Industrial Statistical Yearbook | + |
| | numb | Number of manufacturing enterprise | 2020 Statistical Yearbook | + |
| | Rd | R&D spending divided by main business income | China Industrial Statistical Yearbook | + |

### 4.4.3. Descriptive Statistical Analysis

Before the model regression analysis was performed, the descriptive statistical analysis was performed for the main explanatory variables. This article presents an overall statistical descriptive analysis and the sector data of the Chinese manufacturing industry. Because the stata 15.0 statistical software was adopted, the descriptive statistical results of the relevant variables were obtained as follows in Tables 7 and 8:

**Table 7.** Descriptive statistics of overall manufacturing industry.

| Variables | N | Mean | sd | Min | Max |
|---|---|---|---|---|---|
| $GVC_{pt\_f_{d,t}}$ | 136 | 0.976 | 0.0595 | 0.847 | 1.103 |
| Digin | 136 | 0.0487 | 0.00281 | 0.0445 | 0.0522 |
| Abro | 136 | 1.014 | 0.910 | 0.0595 | 4.228 |
| scal | 136 | 1.91 | 0.612 | 0.0572 | 3.772 |
| Capi | 136 | 2.181 | 0.623 | 1.037 | 3.884 |
| intu | 136 | 0.567 | 0.6231 | 0.0623 | 1.4234 |
| prod | 136 | 1.67 | 0.645 | 0.0545 | 1.2335 |
| numb | 136 | 0.440 | 0.301 | 0.0710 | 1.384 |
| Rd | 127 | 0.00774 | 0.00605 | 0.000785 | 0.0442 |

**Table 8.** Descriptive statistics of sector manufacturing industry.

| Low Knowledge Intensity | | | | | |
|---|---|---|---|---|---|
| Variables | N | Mean | sd | Min | Max |
| $GVC_{pt\_f_{d,t}}$ | 32 | 0.994 | 0.0979 | 0.847 | 1.103 |
| digin | 32 | 0.0487 | 0.00284 | 0.0445 | 0.0522 |
| abro | 32 | 2.032 | 1.085 | 0.309 | 4.228 |
| scal | 32 | 1.953 | 0.326 | 1.243 | 2.106 |
| capi | 32 | 2.059 | 0.306 | 1.353 | 2.506 |
| intu | 32 | 0.592 | 0.174 | 0.182 | 0.923 |
| Prod | 32 | 0.562 | 0.168 | 0.188 | 0.907 |
| numb | 28 | 0.0108 | 0.00510 | 0.00317 | 0.0235 |
| Rd | 32 | 0.994 | 0.0979 | 0.847 | 1.103 |
| Low to medium knowledge intensity | | | | | |
| $GVC_{pt\_f_{d,t}}$ | 40 | 0.970 | 0.0396 | 0.893 | 1.016 |
| digin | 40 | 0.0487 | 0.00283 | 0.0445 | 0.0522 |
| abro | 40 | 0.670 | 0.542 | 0.0595 | 1.660 |
| scal | 40 | 0.593 | 0.516 | 0.0640 | 1.434 |
| capi | 40 | 1.962 | 0.467 | 1.065 | 2.853 |
| intu | 40 | 0.498 | 0.465 | 0.0682 | 1.385 |
| Prod | 40 | 0.572 | 0.603 | 0.0570 | 1.394 |
| numb | 40 | 0.569 | 0.576 | 0.0680 | 1.414 |
| Rd | 40 | 0.469 | 0.476 | 0.0710 | 1.384 |
| Medium and high knowledge intensity | | | | | |
| $GVC_{pt\_f_{d,t}}$ | 64 | 0.970 | 0.0410 | 0.901 | 1.046 |
| digin | 64 | 0.0487 | 0.00282 | 0.0445 | 0.0522 |
| abro | 64 | 0.721 | 0.571 | 0.103 | 2.820 |
| scal | 64 | 0.698 | 0.583 | 0.121 | 1.960 |
| capi | 64 | 2.379 | 0.756 | 1.037 | 3.884 |
| intu | 64 | 0.365 | 0.159 | 0.0704 | 0.625 |
| Prod | 64 | 0.456 | 0.165 | 0.0605 | 0.721 |
| numb | 64 | 0.361 | 0.158 | 0.0806 | 0.621 |
| Rd | 59 | 0.00858 | 0.00723 | 0.000806 | 0.0442 |

The results of descriptive statistical analysis in Table 8 show large differences between different knowledge-intensive manufacturing industries. So, it is necessary to analyze the different sectors of manufacturing industries separately.

### 4.4.4. Unit Root Test

Before applying the fixed effects model, this article uses the LLC test and ADF tests to test the panel data's stationarity to ensure the stability of the data, prevent the occurrence of "pseudo regression", and ensure that the empirical results are more accurate and reliable. Among them, the LLC test applies to the case of a homogeneous unit root, and the ADF applies to the case of a heterogeneous unit root. The test results are shown in Table 9. It can

be seen from the results that all variables passed the LLC test at the 1% significance level and the ADF test. Further testing of various industry data shows that the model data are stable and can be used for further empirical analysis.

**Table 9.** ADF test results.

| Variable/Method | LLC | ADF |
|---|---|---|
| $\text{GVC}_{\text{pt\_f}_{\text{d,t}}}$ | −27.0185 (0.0000) | 154.2681 (0.0000) |
| Digin | −16.4804 (0.0000) | 295.6973 (0.0000) |
| Abro | −8.8498 (0.0000) | 123.0131 (0.0000) |
| Scal | 14.5602 | 235.2906 |
| Cap | −13.9831 (0.0000) | 216.1023 (0.0000) |
| intu | −11.6832 (0.0000) | 208.1023 (0.0000) |
| prod | −12.9831 (0.0000) | 178.903 (0.0000) |
| numb | −10.5270 (0.0000) | 121.0152 (0.0000) |
| Rd | −10.9536 (0.0000) | 71.7653 (0.0000) |

## 5. Empirical Research Results and Analysis

### 5.1. Research Results of the Digital Index (DMI) in the Manufacturing Industry

Table 2 shows that in the 18 sample countries from 2005 to 2017, South Korea's manufacturing industry has the highest degree of digitization, China's manufacturing industry has developed rapidly, and Mexico, Japan, the United States, the United Kingdom, Germany, France, and Sweden have a higher degree of digitization in the overall manufacturing industry. The digitalization of the service industry is high, which is mainly related to the advantages of traditional manufacturing in developed countries [54].

As shown in Figures 3 and 4, from the scale of digital industrialization, the United States is USD 1.5236 trillion, followed by China with USD 968.9 billion. The digital transformation of traditional manufacturing has become the dominant trend. Among them, Germany is the highest, reaching more than 90%. More than 10 countries, including the United Kingdom, the United States, and Russia, also accounted for more than 80%, and China accounted for 79.31%. From the perspective of the level of industrial digitalization, as shown in Figure 4, in 2019, this index and the industry added value accounted for more than 1/3 in South Korea, Germany, the United States, the United Kingdom, and Japan. Among them, South Korea is the highest at 45%. China's industrial digitalization accounts for only 18.3% of the industry's added value, clearly in the initial stage of digital transformation [53].

The results show that the digital level of the manufacturing industry in developed countries is high, often with high industrial added value and strong innovation ability, which leads to its strong competitiveness.

### 5.2. Empirical Results of Structural Equation of the GVC Participation and Location in the Manufacturing Industry

Firstly, the GVC participation and location index, calculated by Equation (2), are shown in Figure 5a. The DVA in China's manufacturing industry has been rising from 2000 to 2017. It indicates that China's manufacturing industry participation in the division of GVC is

increasing. The RDV in China's manufacturing industry is relatively low, and as such, although China's manufacturing industry is large in scale, it is not simultaneously competitive with other economies. The FVA increased about 40%. This shows that the external dependence on China's manufacturing industry's digital transformation was enhanced. The PDC increased, showing that China's manufacturing industry is under pressure from increasing global production costs, economic recession, and trade protection [59].

Secondly, this article used Equations (3) and (4) to analyze the manufacturing GVC location and participation degree in 17 countries. The results show that the manufacturing GVC status index of France, the United Kingdom, Canada, Sweden, Estonia, and Mexico maintained a continuous downward trend from 2000 to 2017. This indicates that the manufacturing industry's international division of labor status in these countries continued to decline. As export-oriented countries, these countries have a relatively high degree of economic dependence on foreign countries and are more vulnerable to the current global economic downturn. The division of labor in the GVC of Germany, Denmark, South Korea, Japan, and the United States has improved, and the GVC status index has shown an upward trend in fluctuations [60].

Thirdly, the empirical results of the GVC participation index in the manufacturing sector are presented. It is shown in Table 6 that from 2000 to 2017, the forward participation of most industries in the manufacturing industry has increased, whereas the backward participation of most industries has declined. In participating in the GVC division of labor, China reduces its dependence on foreign added value and exports domestic added value. The network participation model of some mid-to-high-end manufacturing sectors has gradually shifted from "bottom embedded" to "high-level penetration". China already has a strong independent production capacity in the middle and high-end manufacturing sectors, such as electronics, petrochemicals, and machinery and equipment [61].

The results in Table 6 show the growth of the domestic manufacturing industry; the supporting service industries, such as transportation, finance, and trade have also developed and expanded, showing a trend of increasing forward participation and decreasing backward participation.

From the empirical results, the division of labor status of GVC in the main sample countries shows a stable trend, and the participation shows an increase, indicating that the upgrading of the manufacturing industry chain is accelerating.

*5.3. The Results of Fixed Effects Model Empirical on Digitalization of the GVC of Manufacturing*

The empirical results are shown in Table 8. The coefficient of Digin was also significantly positive, and the t-value did not change significantly. Table 10 shows that after controlling the industry fixed effects, the regression coefficient of Digin is significantly positive, implying the promotion of the status of these sectors in the GVCs. In addition, the regression coefficients of abro and intu in the control variables are all significantly positive. In contrast, the regression coefficients of scal, capi, and prod are significantly negative, which also verifies from the side that China is a manufacturing power. The numb and R&D variables were included for interaction term verification.

In Table 11, eighteen manufacturing sectors in China have been divided into low knowledge intensity (d05, d06, d07, d08, d09 d22), medium and low knowledge density (d10, d11, d12, d13, d14, d15, d16), and medium and high knowledge density (d17, d18, d19, d20, d21), and they have been grouped for empirical analysis. Digital investment has a differentiated effect on the utility of the three groups of the GVC status index. Labor- or resource-intensive sectors with lower technology density negatively affect upgrading their GVC position index. Digital investment has a significant positive effect on upgrading the GVC position index of the other two groups. The medium-high group is stronger than the medium-low group, indicating that the two groups can effectively improve their embedded position in the GVCs by accelerating the digitalization process. In addition, the interaction term models of numb and R&D have negative and positive effects at the 1% significance level for medium and high knowledge intensity groups, respectively [62]. A reasonable

speculation is that the group's R&D investment is more invested in ICT, and the increase in R&D investment is conducive to magnifying this utility.

**Table 10.** Empirical results of benchmark model and the added interaction term.

| Variable Analysis | Reference Model | Numb | R&D |
|---|---|---|---|
| | (1) | (2) | (3) |
| digin | 0.0227 *** | 0.0234 *** | 0.02543 *** |
| | (7.01) | (7.01) | (7.01) |
| abro | 0.0598 *** | 0.0586 *** | 0.0586 *** |
| | (12.95) | (12.55) | (12.01) |
| scal | −0.003 ** | −0.005 ** | −0.004 * |
| | (−3.03) | (−3.19) | (-2.15) |
| cap | −0.0712 *** | −0.0632 *** | −0.0651 *** |
| | (−6.04) | (−6.24) | (−6.57) |
| intu | 0.0263 *** | 0.0244 *** | 0.0236 *** |
| | (6.34) | (6.01) | (6.04) |
| prod | −0.0161 | −0.0169 | −0.0162 |
| | (−1.478) | (−1.483) | (−1.497) |
| numb | | −0.002 | |
| | | (−0.86) | |
| Rd | | | 0.834 |
| | | | (0.40) |
| $a_0$ | 0.231 *** | 0.207 *** | 0.226 *** |
| | (3.49) | (3.18) | (3.36) |
| $R^2$ | 0.158 | 0.146 | 0.147 |
| F | 129.48 *** | 128.22 *** | 125.36 *** |

Notes: ***, **, * indicate significant at the 1%, 5%, and 10% levels, respectively. In parentheses are the t-values.

**Table 11.** The empirical results of groups' measurements.

| Variable Analysis | Low Knowledge Intensity | | | Low to Medium Knowledge Intensity | | | Medium to High Knowledge Intensity | | |
|---|---|---|---|---|---|---|---|---|---|
| | (1) | (2) | (3) | (4) | (5) | (6) | (7) | (8) | (9) |
| digin | −0.001 | −0.004 | −0.004 | 0.027 *** | 0.024 *** | 0.029 *** | 0.038 *** | 0.035 *** | 0.039 *** |
| | (−0.85) | (−0.86) | (−0.79) | (3.54) | (3.65) | (3.15) | (9.74) | (9.76) | (9.01) |
| abro | 0.040 *** | 0.040 *** | 0.041 *** | 0.088 *** | 0.089 *** | 0.089 *** | 0.067 *** | 0.067 *** | 0.066 *** |
| | (7.23) | (7.34) | (7.43) | (5.79) | (5.56) | (5.56) | (11.34) | (11.49) | (11.33) |
| scal | −0.003 * | −0.002 | −0.003 | −0.005 | −0.007 | −0.007 | −0.001 | −0.005 * | −0.002 |
| | (−1.45) | (−0.65) | (−0.09) | (−1.87) | (−1.88) | (−1.85) | (−0.53) | (−1.57) | (−0.64) |
| cap | −0.023 * | −0.024 * | −0.023 * | −0.069 | −0.068 | −0.069 | −0.071 *** | −0.071 *** | −0.072 *** |
| | (−1.96) | (−1.94) | (−1.90) | (−1.68) | (−1.61) | (−1.59) | (−7.11) | (−6.92) | (−6.23) |
| intu | 0.022 *** | 0.022 *** | 0.022 *** | 0.022 ** | 0.022 ** | 0.022 ** | 0.036 *** | 0.036 *** | 0.037 *** |
| | (3.78) | (3.78) | (3.78) | (2.52) | (2.54) | (2.54) | (6.11) | (6.13) | (6.13) |
| prod | 0.002 | 0.002 | 0.002 | −0.04 | −0.036 | −0.037 | −0.026 * | −0.027 ** | −0.027 * |
| | (0.15) | (0.16) | (0.17) | (−0.92) | (−0.96) | (−0.94) | (−1.94) | (−1.97) | (−1.95) |
| numb | | 0.002 | | | −0.003 | | | −0.005 * | |
| | | (0.44) | | | (−0.87) | | | (−1.76) | |
| rede | | | 0.019 | | | 0.047 | | | 0.064 * |
| | | | (3.47) | | | (5.34) | | | (9.76) |
| β0 | 0.009 | 0.016 | 0.010 | 0.211 | 0.218 | 0.218 | 0.232 *** | 0.216 *** | 0.221 *** |
| | (0.06) | (0.14) | (0.19) | (0.79) | (0.84) | (0.75) | (3.87) | (3.66) | (3.43) |
| R2 | 0.1214 | 0.1279 | 0.1464 | 0.3158 | 0.4287 | 0.2224 | 0.7553 | 0.6619 | 0.6652 |
| t/d | | | | | | | | | |
| N | 682 | | | 420 | | | 1012 | | |
| F | 58.14 *** | 58.57 *** | 50.43 *** | 9.49 *** | 9.50 *** | 9.52 *** | 82.54 *** | 82.74 *** | 81.38 *** |

Notes: ***, **, * indicate significant at the 1%, 5%, and 10% levels, respectively. In parentheses are the t-values.

Robustness test as follows:

First, Hummels et al. proposed ($VSI_{d,t}$) as an index of forward vertical division of specialization, which replaced $GVC_{pt\_f_{d,t}}$ [46]. As used for the added value of the final product in other countries, $VSI_{d,t}$ is an intermediate product with the value of a country or region. It was used in the added value of final goods produced by other countries and is a measure of how embedded an economy is in the global industrial chain.

Secondly, the digin_$w_{d,t}$ and digin_$p_{d,t}$ of two variables were used to replace digin_$_{d,t}$. The digin_$w_{d,t}$ shows the added value of worldwide communications and information services used by sector d at time t, and digin_$p_{d,t}$ shows the contribution rate of the domestic digital input to the total output of the department.

Finally, the method of lagging phase was used to test the empirical results. The results show that variables significance and the positive and negative sign of the regression coefficient did not change.

The results show that the promotion effect of digitization on GVC in various sectors of China's manufacturing industry is differentiated. From the perspective of labor-intensive and capital-intensive manufacturing sectors, digital investment increased by 1%, and its GVC division status increased by 0.027% and 0.038%, respectively. The effect on the manufacturing sector with strong technological innovation ability is more significant. It shows that the high-tech manufacturing sector has more competitive advantages, and the digital integration effect is the most obvious. These industries that have mastered the core technology should strengthen China's brand building, enhance the voice of international cooperation, enhance the core competitiveness of China's manufacturing industry, and break down the barriers to entering the high-end market.

## 6. Implications and Suggestions

### 6.1. Implications of Research Conclusions

Research results strongly suggest that China's manufacturing industry should be pushed toward the middle and high end of the GVCs. Facets include:

(1) The comparative analysis of the digital level of the manufacturing industry in 18 key countries shows that although the scale of China's digital economy has been at the forefront of the world, there is a great imbalance between digital development governance, integration, and manufacturing development. China should speed up the construction of digital infrastructure, strengthen the effectiveness of production process control based on 5G and industrial Internet of things platform, tap the hidden potential of data, and speed up the digitization of the manufacturing industry.

(2) The comparative analysis of the GVCs' division status of the manufacturing industry in 18 key countries shows that the GVCs' division status of the manufacturing industry in France, Britain, Canada, Sweden, Estonia, and Mexico continues to decline. As export-oriented countries, these countries are relatively dependent on foreign economies and are more vulnerable to the current global economic recession. This should enlighten China to further improve the industrial chain and reduce the risk of "chain-breaking". The results show that China's manufacturing industry is downstream of GVC, and its export depends on the processing trade industry with low added value. Compared with the United States and Japan, China's manufacturing industry lacks core competitiveness. China should improve the industrial chain further and reduce the risk of "chain-breaking". In addition, China should speed up technological innovation and enhance product competitiveness.

(3) The network participation mode of some medium and high-end manufacturing industries in China has gradually changed from "bottom embedding" to "high-level penetration". China has a strong independent production capacity in the middle and high-end manufacturing industries, such as electronics, petrochemical, and mechanical equipment. China should continue to develop strategic emerging industries and transform and upgrade traditional industries.

(4) The regression analysis of the digital economy and GVC in China's manufacturing industry shows that digitization negatively impacts the improvement of GVC in the labor-intensive or resource-intensive manufacturing industry. Digitization has a positive impact on the improvement of GVC in capital-intensive and technology-intensive manufacturing industries. The digitalization of the manufacturing industry has restrained the advantage of low labor costs in China's manufacturing industry. It enlightens China's labor-intensive or resource-intensive manufacturing industry to accelerate the transformation and upgrading, optimize the structure, and vigorously develop the advanced manufacturing industry. Second, capital-intensive and technology-intensive manufacturing industries can effectively improve their GVC division of labor status by accelerating the digitization process. China's capital-intensive and technology-intensive manufacturing industry should speed up digital construction.

*6.2. Recommendations*

6.2.1. Accelerate the Improvement of China's Digital Quality and In-Depth Construction

Firstly, independent innovation in digital technology should be strengthened and breakthroughs in basic and universal technologies should be accelerated. The innovation ability of key software and hardware technologies should be improved and there should be a focus on fighting the battle of the key core technologies. Secondly, there should be a focus on the in-depth development of cutting-edge digital technologies, such as artificial intelligence, blockchain, Internet of things, 5G, robotics and data mining, and various industries' integrated application. Sustained momentum for China's industrial transformation and upgrading should be provided. Thirdly, the construction of digital infrastructure should be strengthened and the upgrading of network systems and the innovation of basic information facilities should be sped up. Researchers should strive to build a digital ecosystem integrating sensing, transmission, storage, computing, and processing.

6.2.2. Improve the Domestic Industrial Chain and Promote the Balanced Development of Industries

Firstly, the project of "strengthening the chain and supplementing the chain" of the industrial chain and strengthening the domestic economic cycle should be implemented. A multi-circulation system of domestic and international industrial chain should be built, to further supplement and improve China's manufacturing industrial chain and reduce the risk of "chain-breaking". Secondly, a high-level manufacturing technology innovation platform according to the advanced manufacturing industry's major key and common technologies should be built. There should be a focus on breaking through several core technologies that restrict the promotion of GVC in China's manufacturing industry to enhance the competitiveness of China's manufacturing products. Thirdly, China should continue to adjust and optimize the industrial structure of the manufacturing industry, consolidate advantageous industries, and vigorously cultivate and transform the traditional industries. Promote the overall improvement of the value-added capacity of China's export products and realize the manufacturing industry to move forward from the upper reaches of the value chain with low added value to high added value. Fourthly, affected by low labor costs in developing countries such as Brazil and India, the competition at the low end of the industrial chain is becoming increasingly fierce. The profit margin of China's low-end manufacturing industry continues to decline, and the bottom of the "smile curve" continues to sink. When developed countries dominate the high end of the GVCs, to upgrade from ODM to OBM, China should actively expand domestic and international markets, occupy the domestic sales market, pay attention to industrial upgrading, scientific, and technological innovation, consolidate the foundation of high-end manufacturing industry, and enhance its core competitiveness, to break through the low end of the GVCs.

### 6.2.3. Driven by the Digital Economy, Promote GVC in China's Manufacturing Industry

Firstly, the incentive policies should be improved. The digital transformation of enterprises should be encouraged and the transformation and upgrading of enterprises through technology loan projects, subsidy or interest, and industrial guidance fund investment should be realized. Secondly, the integration with international manufacturing digital standards should be promoted. The forward-looking layout of digital technology should be promoted and the core and key technologies of digital production in the manufacturing industry should be mastered. Thirdly, a good manufacturing digital innovation environment should be created. A legal system for the development of the digital manufacturing economy should be established. An ecosystem of manufacturing digitization, such as co-construction and sharing, information security and intellectual property, protection should be built. Fourthly, high-level and sophisticated talents in digital technology in the manufacturing industry should be vigorously introduced and facilities should be well supported, such as capital and housing subsidies. At the same time, the reward and punishment mechanism should be improved and several discipline leaders related to the digital construction of the manufacturing industry should be cultivated. Fifthly, because of the normalization of the prevention and control of COVID-19 in China, China should accelerate the digitalization of its manufacturing industry. It should give full play to the advantages of data aggregation, resource scheduling, and data analysis to improve the emergency response efficiency in the epidemic. Sixthly, manufacturing enterprises should be helped to move their sales offline to online and accurately connect with customers. Then, manufacturing enterprises should be helped to locate upstream suppliers quickly and improve raw material procurement efficiency. Finally, enterprise remote office support should be provided to reduce the frequency of front-line personnel going to and from the factory.

Due to the limitations of the quantitative analyses of this study, the method itself needs further improvement. Specifically, future researchers can accelerate investigations of the digital economy's multi-dimensional evaluation index systems. Future research will further examine the digital infrastructure, R&D input, talents, digital technology, and digital governance indicators. The authors will further research digital technology's effect on China's labor-intensive, capital-intensive, and technology-intensive manufacturing industry and focus on the effect of value creation of the GVC on each link of the space network layout. Future research will also involve deepening the research on the integration of digital technology and manufacturing R&D investment and the effect of globalization. Moreover, it will involve realizing cross-border links in the GVC of China's manufacturing industry under the international vertical division of the labor system. Additionally, future research will still involve strengthening the research on the impact of blockchain technology on mutual trust [63], sustainable supply chain [64], and enterprise value chain system reconstruction based on blockchain technology in the accounting domain [65]. This will help to realize the sustainable development of China's manufacturing industry and encourage the promotion of the GVCs.

**Author Contributions:** Conceptualization, R.Z. and D.T.; methodology, R.Z.; software, L.K. and V.B; validation, R.Z., W.C., D.T. and V.B.; formal analysis, R.Z. and W.C.; investigation, L.K. and D.D.; resources, R.Z.; data curation, R.Z.; writing—original draft preparation, R.Z.; writing—review and editing, R.Z., D.T., D.D. and V.B.; visualization, W.C., R.Z., D.T., D.D. and V.B.; supervision, D.T. and D.D.; project administration, R.Z.; funding acquisition, R.Z. All authors have read and agreed to the published version of the manuscript.

**Funding:** General Project of National Social Science Foundation of China "Research on China's Manufacturing Industry Moves to the Middle and High-end Value Chain under the New Development Pattern of Dual Cycle Driven by Digital Economy" (No. 21BJY085); China Statistical Science Research Project "Effect Evaluation and Countermeasures of Advanced Manufacturing Industry Cluster in China" From National Statistics Bureau of China (2020LY106); Jiangsu Province Policy Guidance Soft Science Project "Digital Economy Drives Manufacturing in Jiangsu Research on the industry's strategy moving towards the middle and high end of the value chain" (BR2021004).

**Institutional Review Board Statement:** Not applicable.

**Informed Consent Statement:** Not applicable.

**Data Availability Statement:** All Data available in "World Input-Output Database (WIOD)" ([https://www.wiod.org](https://www.wiod.org), accessed on 10 April 2022), TIVA Database [[https://www.apec.org](https://www.apec.org) ([scyky.com](scyky.com)), accessed on 10 April 2022], ADB Database ([https://docs.oracle.com/](https://docs.oracle.com/), accessed on 10 April 2022) and National Bureau of Statistics ([https://www.stats.gov.cn](https://www.stats.gov.cn), accessed on 10 April 2022).

**Conflicts of Interest:** The authors declare no conflict of interest.

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
