# Peer review of "Research on China’s Manufacturing Industry Moving towards the Middle and High-End of the GVC Driven by Digital Economy"

_sustainability, doi:10.3390/su14137717_

Round 1

Reviewer 1 Report

I am happy for reviewing your manuscript which was very interesting to me. It is certainly an interesting topic, particularly for developing countries and emerging markets that are under pressure to balance their rapid economic growth with enhance core competitiveness. Some of the issues that need to be addressed are as follows.

1.Table 1 is better used instead of figures. It is rare to use tables in the introduction section for describing the content in general.

2.What does the " a high-end fixed effect " indicate?

  1. Is the measurement method of Digitization of Manufacturing Index (DMI) credible? We know that ICT does not represent all digital economy industries, and ICT-related industries should be included.
  2. The literature review is not well summarized. Authors have simply described previous studies.
  3. The introduction needs to be rewritten to improve research gaps this paper.

6.Discussing the results is ignored. A good paper should explain the economic phenomenon behind the empirical results

7.Robustness tests should be performed to ensure the validity of the empirical results.

  1. Please explain why the research in this paper meets the scope of this journal's selection.

Author Response

Dear reviewer, thank you very much for your valuable comments and suggestions. According to your proposal, we have made a substantial revision of the paper from several aspects, such as the introduction, literature review and the final discussion, so as to get your approval.

Point 1: Table 1 is better used instead of figures. It is rare to use tables in the introduction section for describing the content in general.

Response 1: According to your suggestions, we deleted Table 1 and described it directly with figures. Please see Line 56-72

Point 2: What does the " a high-end fixed effect " indicate?

Response 2: We are very sorry. Here is our expression error. It should be fixed effects model. We have modified it and checked the full text.

Point 3: Is the measurement method of Digitization of Manufacturing Index (DMI) credible? We know that ICT does not represent all digital economy industries, and ICT-related industries should be included.

Response 3: According to your suggestions, we discussed this method again. We found that the United Nations Organization for economic cooperation and development, the National Bureau of statistics of China, Mc Kinsey and other institutions all use ICT as the core industry of digital economy and carry out digital computing. In addition, we have also considered your questions. However, due to the unclear boundary of ICT related industries, this problem has not been well solved in the field of digital measurement. We will make a detailed study on this problem later.

Point 4: The literature review is not well summarized. Authors have simply described previous studies.

Response 4: According to your suggestions, we summarize the literature review again. Please see Line328-341

Point 5: The introduction needs to be rewritten to improve research gaps this paper.

Response 5: According to your suggestions, we rewrite the introduction to improve research gaps of this paper. Please see Line 84-106

Point 6: Discussing the results is ignored. A good paper should explain the economic phenomenon behind the empirical results

Response 6: According to your suggestion, we have revised several parts of the paper. Please see Line 714-716, 753-755, 774-785

Point 7: Robustness tests should be performed to ensure the validity of the empirical results.

Response 7: According to your suggestions, we have added stability test. Please see Line 682-693

Point 8: Please explain why the research in this paper meets the scope of this journal's selection.

Response 8: This paper studies the promotion of the GVCs in China's manufacturing industry around the digital economy. By accelerating the development of digital economy and improving the GVC division status of China's manufacturing industry, it will help to promote the full utilization of labour, resources, and other production factors in China's manufacturing industry, to reduce the waste of production factors and promote the sustainable development of China's manufacturing industry.

Reviewer 2 Report

Article: Research on the Effect of the Promotion of China’s Manufacturing Global Value Chain Driven by the Digital Economy

Is the content succinctly described and contextualized with respect to previous and present theoretical background and empirical research (if applicable) on the topic?

Partially - The title states “effect of the promotion of China’s manufacturing GVC driven by the digital economy” but the abstract doesn’t point out what kind of “effect”. In the abstract, the authors state that digital economy “can enhance independent innovation efficiencies, promote the development of advanced manufacturing clusters and constantly spawn new models, forms of business and industries”, at the same time, employ variables representing digital economy, so it can be understood that the authors try to point out the relationship between digital contents in manufacturing sectors and participation of those sectors in the - as stated “due to insufficient R&D investment in the division of labour in the GVC, China's manufacturing industry is prone to low-end lock-in, inefficient industrial  structures, and weak innovation ability” and recommended on promoting manufacturing GVC to “move up to the middle and high end along the GVC”. The title should be revised to clarify the “effect”, which the authors want to convey through this empirical study.

Also, the authors need to review more existing relevant publications. In literature review sections:

+ Section 2.1 only reviews “digital economy” and “GVC theory” separately without pointing out the relationship between digital economy and GVC theory.

+ Section 2.2 just mentions that between trade and level of GVC participation and doesn’t show the relationship between “division of labour” and “level of GVC participation” (it should clarify how division of labour is impacted/creating impact from patterns of GVC).

+ Sub-section 2.3.2 should be replaced in the literature review on GVC and clarify on how existing study (models, methods...) examined the relationship between digitalization and GVC.

+ Sub-section 2.3.3 should clarify pros and cons of each method and how each method can be applied in to empirical study.

+ Sub-section 4.4.2 mentions variables used by the authors so the literature review should mention about how existing study employed those variables and clarify why the variables are selected

Are all the cited references relevant to the research?

Yes - But the authors need to provide more references for statements and statistical quotes (for example, line 37-46, 49-50, 50-57, 66-74, 158-168, 194-195, 318-333, 337-344, 362-372, 467-477, 479-495... “12 consecutive years” in line 50 or “Machlup” in line 254, direct quoting term “troika” in line 158...) as well as sources and years in figures and tables in the article

Are the research design, questions, hypotheses and methods clearly stated?

Yes - But the authors should clarify more about the variables representing digitalization/digital economy in sub-section 4.4.2 by providing more literature review on existing study on relevant variables. Also, the data mentioned in the article should be reconsidered as the data on GVC is only up to 2017, on digitalization is up to 2018 but on market scale and growth rate is during 2018 - 2020 (is the data compatible with the context and purpose of this article to support the authors’ statements?).

Are the arguments and discussion of findings coherent, balanced and compelling?

Partially - The authors should clarify the purpose of selecting 18 countries for calculation and how the study of 18 countries will provide implications on “the effect of the promotion of China’s manufacturing global value chain driven by the digital economy”.

For empirical research, are the results clearly presented?

Yes - But the authors should provide more details on the reliability of the calculating outcomes.

Is the article adequately referenced?

Partially - it needs to be improved with more specified literature review (as mentioned above in comments on literature review).

Are the conclusions thoroughly supported by the results presented in the article or referenced in secondary literature?

Partially - The authors need to conclude on how the study of 18 countries will provide implications for the case of China. Also, the authors mentioned “the outbreak and spread of COVID-19 pandemic” as a context factor for “the acceleration of the global industrial chain reconstruction”, so the authors should provide more implications and recommendations for China on how digitalization could support China to accelerate its manufacturing sectors in the context that China’s manufacturing GVC has been disrupted by the pandemic.

Author Response

Dear reviewer, thank you very much for your valuable comments and suggestions. According to your proposal, we have made a substantial revision of the paper from several aspects, such as the title, literature review and suggestions, so as to get your approval.

Point 1: Is the content succinctly described and contextualized with respect to previous and present theoretical background and empirical research (if applicable) on the topic?

Response 1: According to your suggestions, we discussed the theme of the article according to the content of the article. Finally, in order to make the content of the article more relevant to the theme, we decided to revise the title of the article as: Research on China's manufacturing industry moving towards the middle and high-end of the global value chains driven by digital economy.

Point 2: The title states “effect of the promotion of China’s manufacturing GVC driven by the digital economy” but the abstract doesn’t point out what kind of “effect”. The title should be revised to clarify the “effect”, which the authors want to convey through this empirical study.

Response 2: According to your suggestions, we decided to revise the title of the article as: Research on China's manufacturing industry moving towards the middle and high-end of the global value chains driven by digital economy.

Point 3: Section 2.1 only reviews “digital economy” and “GVC theory” separately without pointing out the relationship between digital economy and GVC theory.

Response 3: Thank you for your suggestions. The literature review on the relationship between digital economy and GVC is in line 241-254 of the article.

Point 4: Section 2.2 just mentions that between trade and level of GVC participation and doesn’t show the relationship between “division of labour” and “level of GVC participation” (it should clarify how division of labour is impacted/creating impact from patterns of GVC).

Response 4: According to your suggestions, we have added  the relationship between "division of labour" and " level of GVC participation ". Please see Line 200-209

Point 5: Sub-section 2.3.2 should be replaced in the literature review on GVC and clarify on how existing study (models, methods...) examined the relationship between digitalization and GVC.

Response 5: Thank you very much for your suggestions. Sub section 2.3.2 is a summary of the calculation method of GVC division of labour. This paper uses the decomposition method of export trade added value to calculate the GVC division of manufacturing industry in 18 countries including China. To compare and analyse the current situation of GVC in China's manufacturing industry. Moreover, this paper uses the fixed effect regression model to study the interaction between China's digitization and China's manufacturing GVC. Taking the GVC division status of China's manufacturing industry as the explained variable and digitization as the explanatory variable. Therefore, we do not recommend replacement. In addition, there are few literatures on Digitization and GVC. It also focuses on the theoretical research on the division of labour status of digital global value chain, with less quantitative research and less relevant models or methods. In this paper, the fixed effects model is used to study the relationship between digitization and manufacturing GVC, which is a supplement to the missing research content.

Point 6: Sub-section 2.3.3 should clarify pros and cons of each method and how each method can be applied in to empirical study.

Response 6: According to your suggestions, we clarified the advantages and disadvantages of each method and how to apply each method to empirical research. Please see Line 309-319.

Point 7: Sub-section 4.4.2 mentions variables used by the authors so the literature review should mention about how existing study employed those variables and clarify why the variables are selected.

Response 7: According to your suggestions, we add relevant literature on the variables used in sub-section 4.4.2 in the summary of literature review sub-section 2.2. Please see Line 220-231. In addition, we clarify the reasons for selecting variables in sub-section 4.4.2. Please see Line 658-672

Point 8: Are all the cited references relevant to the research?

Yes - But the authors need to provide more references for statements and statistical quotes (for example, line 37-46, 49-50, 50-57, 66-74, 158-168, 194-195, 318-333, 337-344, 362-372, 467-477, 479-495... “12 consecutive years” in line 50 or “Machlup” in line 254, direct quoting term “troika” in line 158...) as well as sources and years in figures and tables in the article.

Response 8: According to your suggestions, we have added references in the corresponding places. The new references are the 3rd, 4th, 5th, 7th, 24th, 39th, 49th, 56th, 57th, 61st and 62nd respectively. We also explained the data of figures and tables in the article in the corresponding places.

Point 9: Are the research design, questions, hypotheses and methods clearly stated?

Yes - But the authors should clarify more about the variables representing digitalization/digital economy in sub-section 4.4.2 by providing more literature review on existing study on relevant variables. Also, the data mentioned in the article should be reconsidered as the data on GVC is only up to 2017, on digitalization is up to 2018 but on market scale and growth rate is during 2018 - 2020 (is the data compatible with the context and purpose of this article to support the authors’ statements?).

Response 9: According to your suggestions, we add relevant literature on the variables used in sub-section 4.4.2 in the literature review. Please see Line 220-231. We added to clarify the reasons for selecting variables in sub-section 4.4.2. Please see Line 658-672. Thank you for your suggestions on data. Because this study involves many countries and is affected by data availability constraints, the calculation of division of labour status of global value chain is based on the latest 2017 data of WIOD, TIVA and ADB. Digitalization is the data as of 2018 and the data of market size and growth rate from 2018 to 2020 only involve China. This paper uses this part of data to explain the current situation of China's digital economy. The research on the relationship between data economy and GVC in this paper uses the relevant data of GVC up to 2017. Therefore, the data used in this paper is consistent with the background and purpose of this paper.

Point 10: Are the arguments and discussion of findings coherent, balanced and compelling?

Partially - The authors should clarify the purpose of selecting 18 countries for calculation and how the study of 18 countries will provide implications on “the effect of the promotion of China’s manufacturing global value chain driven by the digital economy”.

Response 10: By calculating the GVC division status of manufacturing industry in 18 countries from 2000 to 2017, China is compared with 17 other countries. It fully shows that the development situation of GVC in China's manufacturing industry is grim and needs to be improved urgently. The 17 countries selected in this paper include developed and developing countries, which are representative. Please see Line 73-83, 790-830

Point 11: For empirical research, are the results clearly presented?

Yes - But the authors should provide more details on the reliability of the calculating outcomes.

Response 11: According to your suggestions, we have added stability test. Please see Line682-693

Point 12: Is the article adequately referenced?

Partially - it needs to be improved with more specified literature review (as mentioned above in comments on literature review).

Response 12: According to your previous suggestions, we have improved the literature review of the article.

Point 13: Are the conclusions thoroughly supported by the results presented in the article or referenced in secondary literature?

Partially - The authors need to conclude on how the study of 18 countries will provide implications for the case of China. Also, the authors mentioned “the outbreak and spread of COVID-19 pandemic” as a context factor for “the acceleration of the global industrial chain reconstruction”, so the authors should provide more implications and recommendations for China on how digitalization could support China to accelerate its manufacturing sectors in the context that China’s manufacturing GVC has been disrupted by the pandemic.

Response 13: According to your suggestions, we summarize how this study can provide enlightenment for China. In combination with the development status of COVID-19, it puts forward more relevant countermeasures and suggestions based on digitalization to promote the promotion of GVC in China's manufacturing industry. Please see Line 882-886

so the authors should provide more implications and recommendations for China on how digitalization could support China to accelerate its manufacturing sectors in the context that China’s manufacturing GVC has been disrupted by the pandemic

Response 13-1: According to your suggestions,We make suggestions:Line 832-902

Reviewer 3 Report

The article is relevant from the perspective of developing countries, which fases the challenge of moving to medium and high end segments in the value chains. The empirical approach is valuable and its findings. The literature discussion is very affortunate. 
Two comments:
1- Line 133, regarding the mention to Porter. Strictly Porter afords the microeconomic concept of value chain. In a more extended way he uses the value system concept, which is closer to Gereffi´s GVC approach. Perhaps it it is neccesary an ajustemenin the writing.
2- The empirical results confirm the trend for developing countries regardins the quality of their participación in value chains. The case of China is ilustrative in that sense because in spite of its policies to promote China´s manufacturing GVC driven by the digital economy, it seems that is not enough to participate in such value chains. Entrance barriers coming from the governance of the industry emerge, and also for China they affect its endevours. In this sense perhaps is neccesary to include some reflections regarding qualitative analysis to clarify the kind of efforts required to handle with the governance of the value chains.

Author Response

Dear reviewer, thank you very much for your valuable comments and suggestions. According to your proposal, we have made a substantial revision of the paper from two aspects, so as to get your approval.

Point 1: Line 133, regarding the mention to Porter. Strictly Porter afords the microeconomic concept of value chain. In a more extended way he uses the value system concept, which is closer to Gereffi´s GVC approach. Perhaps it it is neccesary an ajustemenin the writing.

Response 1: We have improved the GVC regional characteristic theory according to your suggestions. Kogut's research results are added between Porter and Gereffi to improve the development process of GVC theory. GVC goes from the enterprise level within the country to the national level between regions and then to the global level. Please see Line 260-263

Point 2: The empirical results confirm the trend for developing countries regardins the quality of their participación in value chains. The case of China is ilustrative in that sense because in spite of its policies to promote China´s manufacturing GVC driven by the digital economy, it seems that is not enough to participate in such value chains. Entrance barriers coming from the governance of the industry emerge, and also for China they affect its endevours. In this sense perhaps is neccesary to include some reflections regarding qualitative analysis to clarify the kind of efforts required to handle with the governance of the value chains.

Response 2: According to your suggestions, we added relevant research results of value chain governance in the article’s literature review. Please see Line 210-219 Relevant countermeasures are added in the countermeasures part. Please see Line 857-866.

Round 2

Reviewer 1 Report

Many thanks to the authors for the review comments revision. But there are some points that are very confusing to me. Why is the revision uploaded using the reviewing mode, which seems to be very messy. Also, the author needs to provide a clean version.

Author Response

Dear Reviewer,

       Thank you very much for your valuable advice. After the manuscript was changed to the clean version, the line number of the cover letter content changed, so we have proofread the line number again. Please find attached the updated Responsing letter.

This manuscript is a resubmission of an earlier submission. The following is a list of the peer review reports and author responses from that submission.

Round 1

Reviewer 1 Report

The paper deals with an interesting issue and is generally well prepared. The theoretical background is presented, methodology is transparent (however, not justified why it has been selected) and results are described in detail on a descriptive level.

A big issue is, that the paper is missing a justification of the research gaps, a presentation of the research questions (or hypothesis) and the research objectives of this paper. Why is this topic relevant? What are gaps in existing literature? What does you work contribute in terms of closing this gap and advancing the body of knowledge?

The presented scientific background is not analysed in terms of its limitations – what is missing and what are you contributing with you research?

The results are presented rather descriptive, i.e., facts and findings are presented without interpretation and deeper analysis of their relevance to your paper or the research objective of the paper.

In section 5.1., I am missing references for the statements. In general, section 5.1.

Section 5.2. provides valuable recommendations for policy making – is this the main contribution of the paper? What are the research contributions? Hard to understand, as research questions, objectives and gaps are missing. The conclusion section is missing scientific contributions of your paper – how do your results promote academia? What are the limitations of the paper? How can your results provide an avenue for future research? This is a major gap in your paper.

You mention the effects of Covid-19 a couple of times, but I am missing a more detailed explanation of these effects in the context of your paper. What difference did it make, how did the pandemic influence your results – what are pandemic-related explanations of expected or unexpected findings.

In my opinion, the paper needs significant and major revisions in terms of i) justifying the relevance of you topic form a scientific point of view, ii) stating clear research objectives and corresponding research questions or hypothesis, iii) providing in-context analysis and deeper analysis rather than mainly descriptive presentation of results, iv) discussing your results against the existing body of knowledge – what are new / contradicting / confirming results?, v) what are the limitations of your study? How serious are they and how could future research overcome these limitations? and vi) what is the impact of your results for future research in general – what avenues arise, what additional research needs may be observable?

General formatting, citation, and grammar issues:

  • The labels of figure 1 are not entirely displayed, please adapt.
  • Some sentences are hard to comprehend and very long (e.g., “With data as the key element, industrial Internet, intelligent manufacturing, Internet of Vehicles and other new integrated industries and new models as the main content, with value release as the core and empowerment, the new generation of digital technologies such as data integration, platform empowerment, etc. lead the entire industry chain Upgrade and transform, and carry out the all-round, multi-angle and full-chain trans-formation of the traditional manufacturing industry.”). I would suggest English proofreading and shortening of too complex sentences.
  • Please make sure to use citation consistently. (e.g., here: “As shown in Figure 2, according to CCW Research's "2019-2020 China Key Industry Digital Transformation Market Status and Development Trend Research Report", due to the impact of Covid-19, the growth rate of digital transformation of the manufacturing industry is under increasing pressure, with a year-on-year increase of 9.3%). In this section as well as in figure 2 itself, no citations are available, except the in-text mentioning of CCW Research. Please apply the same citation style consistently. For the entire section 3.2., it is unclear where the comparative values come from – what is the source? Please add citations.

Author Response

Point 1. Justifying the relevance of you topic form a scientific point of view.

Response 1:Please see Page 1(line 31-34)and Page 2(line 43-47).

We have revised it as follows:

Since 2020, the COVID-19 pandemic has spread globally, it has been accelerating the reconstruction of the global industrial chain. Now, the global industrial chain and supply chain are facing a trend of shortening, and regionalization has become an important way of economic cooperation [1].(line 31-34)

The digital economy is an emerging economic form based on the three elements of big data, intelligent algorithm and computing power platform. It is a key measure to promote the innovation level of China's manufacturing industry, get rid of the path dependence of "low-end lock", and realize the high-end of the value chain.(line 43-47)

Point 2. stating clear research objectives and corresponding research questions orhypothesis, a justification of the research gaps, a presentation of the research questions (or hypothesis) and the research objectives of this paper. Why is this topic relevant? What are gaps in existing literature? What does you work contribute in terms of closing this gap and advancing the body of knowledge?

Response 2:Please see “2.3.5.Research Aim and Research QuestionsMake up the gap” (Page 6 to Page 7: Line 281-301)

Point 3. Providing in-context analysis and deeper analysis rather than mainly descriptive presentation of results.

Response 3:

We have made extensive additions and revisions to this. See article 4.5 to 4.5.4: pages 20 to 22 (lines 651 to 688)

Point 4. Discussing your results against the existing body of knowledge – what are new / contradicting / confirming results?,

Response 4:

Please see 4.6.1-4.6.4 (Page 21: Line 690-735)

Point 5. What are the limitations of your study? How serious are they and how could future research overcome these limitations?

Response 5:Please see “4.6.4 Limitations of the research” (Line 723-735, P. 22)

Point 6. What is the impact of your results for future research in general – what avenues arise, what additional research needs may be observable?

Response 6:Please see Line 728-735, Page 22.

In the future, this paper will strengthen the research on the logical mechanism of the impact of the digital industry on the global value chain improvement of the manufacturing industry and the supporting index system of the digital level determination, and further analyze the impact of the digital economy on the status and length of the global value chain of the manufacturing industry.

Enhance the comprehensive use of WIOD, ADB, and RECD databases, and increase the number of world sample analysis countries in order to improve the accuracy and representativeness of the research results.

Point 7.In section 5.1., I am missing references for the statements. In general, section 5.1.

Response 7:In section 5.1.added references for the statements, line729-756, Page 22-23.

Point 8. Section 5.2. provides valuable recommendations for policy making – is this the main contribution of the paper? What are the research contributions? Hard to understand, as research questions, objectives and gaps are missing. The conclusion section is missing scientific contributions of your paper – how do your results promote academia? What are the limitations of the paper? How can your results provide an avenue for future research? This is a major gap in your paper.

Response 8: We have carefully revised the article according to the comments of reviewers. For details, Please see: 4.5-4.6, lines 653-698, Page 20-21.

Point 9. You mention the effects of Covid-19 a couple of times, but I am missing a more detailed explanation of these effects in the context of your paper. What difference did it make, how did the pandemic influence your results – what are pandemic-related explanations of expected or unexpected findings.

Response 8: Please see Line 740-744, Page 22.

Affected by Covid-19, the sharp decline in production and consumption demand not only highlights the huge risk of the world economic recession, but also profoundly affects the GVC, industrial chain and supply chain, which means the intensified industrial regionalization and localization. At the same time, the market's enthusiasm for innovation has been stimulated, and a series of new services, new forms of business and new models have emerged. It obviously presents the development trend of industrial platform, digitalization and intelligence.[54].

General formatting, citation, and grammar issues:

  1. The labels of figure 1 are not entirely displayed, please adapt.

Response A:It has been revised.

  1. .Some sentences are hard to comprehend and very long (e.g., “With data as the key element, industrial Internet, intelligent manufacturing, Internet of Vehicles and other new integrated industries and new models as the main content, with value release as the core and empowerment, the new generation of digital technologies such as data integration, platform empowerment, etc. lead the entire industry chain Upgrade and transform, and carry out the all-round, multi-angle and full-chain trans-formation of the traditional manufacturing industry.”). I would suggest English proofreading and shortening of too complex sentences.

Response B:We revised them and some other sentences in the article. Please see line 368-380, page 9-10.

  1. Please make sure to use citation consistently. (e.g., here: “As shown in Figure 2, according to CCW Research's "2019-2020 China Key Industry Digital Transformation Market Status and Development Trend Research Report", due to the impact of Covid-19, the growth rate of digital transformation of the manufacturing industry is under increasing pressure, with a year-on-year increase of 9.3%). In this section as well as in figure 2 itself, no citations are available, except the in-text mentioning of CCW Research. Please apply the same citation style consistently. For the entire section 3.2., it is unclear where the comparative values come from – what is the source? Please add citations.

Response C:

“As shown in Figure 2, according to CCW Research's "2019-2020 China Key Industry Digital Transformation Market Status and Development Trend Research Report".

It was revised to :

As shown in Figure 2, according to The Global Industry Research Institute of Qinghua University's "Research Report on the Digital Transformation of Chinese Enterprises (2020)"[39]

“due to the impact of Covid-19, the growth rate of digital transformation of the manufacturing industry is under increasing pressure, with a year-on-year increase of 9.3%). In this section as well as in figure 2 itself, no citations are available”

It was revised to :

according to The Global Industry Research Institute of Tsinghua University's "Research Report on the Digital Transformation of Chinese Enterprises (2020)"[39], due to the impact of Covid-19, the growth rate of digital transformation of the manufacturing industry is under increasing pressure[54], with a year-on-year increase of 9.3%. The market size has reached 245.5 billion yuan.[39]

“For the entire section 3.2., it is unclear where the comparative values come from – what is the source? Please add citations.”

We added citations: section 3.2 (line402-425, Page 10-11)

Reviewer 2 Report

Overall, the paper "Research on the Effect of the Promotion of China's Manufacturing Global Value Chain Driven by the Digital Economy" presents some fascinating ideas. Authors have developed the paper logically and coherently. The writing style is concise and structured well. However, I have a few concerns which I would like to put forward to improve the quality of the paper.

  1. Authors need to clarify the research gaps and how the present study enables bridging those gaps. It would be better to give some references to support this study. Obviously, the authors did not do this very well.
  2. The research background should be extended to include relevance, motivation, importance, uniqueness, and contribution to the literature.
  3. More hot literature on the digital economy should be cited.
  4. The content of the literature review also failed to cover most of the previous studies, at least the selected literature should be representative.
  5. Results should be discussed.
  6. Research limitations and outlook should be given.

Author Response

  1. Authors need to clarify the research gaps and how the present study enables bridging those gaps. It would be better to give some references to support this study. Obviously, the authors did not do this very well.

Response 1:Please see Line281-301, Page 6-7.

2.3.5.Research Aim and Research Questions、Make up the gap

On the basis of existing literature research, the gap for a scientific definition of the digital economy, quantitative analysis, the application of scientific division of labor of manufacturing global value chain, and research on how to accelerate the competitiveness of the global value chain of China's manufacturing chain.

This paper theoretically defines the scientific connotation of the digital economy, which drives the logical mechanism for the manufacturing industry to be more embedded in the global value chain, participate in the international division of labor, and obtain more competitive interests. The digital index is used to determine the digital level of the manufacturing industry in China and major countries for comparative analysis; The WWYZ structural equation is used to determine the position of the Chinese manufacturing industry in the global value chain. At the same time, the introduction of digital level, research and development investment, export added value, manufacturing added value scale, and other variables of the influence on the division of labor in the manufacturing global value chain was analyzed. The research results support: (1) Developed countries have a high level of digitalization and a leading position in the global value chain. (2) China's all manufacturing industry is at the low end of the global value chain. The level of digitalization is relatively low, showing the overall trend of accelerated development. (3) We shall accelerate the development of the digital economy, raise the capacity for independent innovation, and realize the transformation and upgrading of China's manufacturing industry.(line381-301)

  1. The research background should be extended to include relevance, motivation, importance, uniqueness, and contribution to the literature.

Response 2:Please see Line 51-71, Page 2.

From the theoretical level, this paper analyzes the problem of China's manufacturing industry being "big but not strong", and the overall manufacturing industry is still in the middle and low end of the global value chain. From the perspective of the imbalance of interest distribution under the global value chain system, how much a country benefits from economic globalization depends on how much it is embedded into the global value chain system. Developed countries occupy high value in the global value chain by providing core components to other countries, while most developing countries are squeezed into the low-added value processing and assembly link. Compared with other manufacturing powers, China's manufacturing products are easier to be replaced, resulting in weak international competitiveness. As a result, China is in urgent need of realizing the transformation and upgrading of its manufacturing industry and climbing to the medium-high end along the global value chain. This paper, from the perspective of digital manufacturing industry, puts forward the digital technology represented by the Internet, big data, cloud computing and the Internet of Things, which accelerate the change. Through the fuse with the traditional manufacturing industry, we can realize the profound change of organizational production mode and business model, so as to integrate and reconstruct the global value chain at a faster speed and a wider range.

This paper is exactly from the perspective of the digital economy, theoretical and quantitative analysis of digital effect on the division of labor in manufacturing global value chain, it has theoretical and practical significance.(line 51-71)

  1. More hot literature on the digital economy should be cited.

Response 3:We added digital economy’s references to the article:

[5] Xiaoxia Chen , Mélanie Despeisse and Björn Johansson.Environmental Sustainability of Digitalization in Manufacturing: A Review.Sustainability 2020, 12, 10298;  2 of 31 .doi:10.3390/su122410298

  • Nascimento, D.L.M.; Alencastro, V.; Quelhas, O.L.G.; Caiado, R.G.G.; Garza-Reyes, J.
  • ; Rocha-Lona, L.; Tortorella, G. Exploring Industry 4.0 technologies to enable circular economy practices in a manufacturing context: A business model proposal. J. Manuf. Technol. Manag. 2019, 30, 607–627. [CrossRef]

[8] Legner, C.; Eymann, T.; Hess, T.; Matt, C.; Böhmann, T.; Drews, P.; Mädche, A.; Urbach, N.; Ahlemann, F. Digitalization: Opportunity and Challenge for the Business and Information Systems Engineering Community. Bus. Inf. Syst. Eng. 2017, 59, 301–308. [CrossRef]

[9]Yoo, Y.; Lyytinen, K.; Thummadi, V.; Weiss, A. Unbounded Innovation with Digitalization: A Case of Digital Camera. In Proceedings of the Annual Meeting of the Academy of Management Montréal, Montréal, QC, Canada, 6–10 August 2010; pp. 1–41

[10]Porter, M.E. Competitive Advantage. New York, Free Press 1985.

[11] Arushanyan, Y.; Ekener-Petersen, E.; Finnveden, G. Lessons learned—Review of LCAs for ICT products and

[12]services. Comput. Ind. 2014, 65, 211–234. [CrossRef]

[44]A moako, T., Sheng, Z.H., Dogbe, C.S.K., Pomegbe, W.W.K., 2020. Effect of internal integration on SMEs’ performance: the role of external integration and ICT. Int. J.

Prod. Perf. Manag. https://doi.org/10.1108/IJPPM-03-2020-0120 ahead-of-print.

  1. The content of the literature review also failed to cover most of the previous studies, at least the selected literature should be representative.

Response 4:Please see Line 194-219, Page 5.

British evolution economist Carota Perez [51] clearly stated that "every technological revolution forms a suitable technological and economic paradigm". At present, it is in the changing period of iterative new technology and economic paradigm. The concept of global value chain proposed by Michael Porter in 1985 provides a brand new dimension for evaluating the industrial advantages and enterprise competitiveness of an economy. This definition covers the core modules of power system, governance structure and industrial upgrading.

Later, Gereffi et al. [12] used the 3C model to observe the dynamic changes of industrial structure. Kaplinsky [46] proposed "inter-industry value chain and intra-industry value chain" and "chain horizontal governance" industrial cluster from the perspective of endogenous complementarity, which continuously broadened the research boundary. However, Bell believed that it is rare to make a complete conclusion on the industrial upgrading of the value chain from any aspect.

More consensus is reached that practice has proved that developing countries are locked in low value-added links with low barriers in the global value chain, and the fierce competition causes the dilemma of "poverty growth", which will not lead to technological progress [16].

Obviously, as a great power strategy, the sequential upgrading process of process, product, function and chain proposed by .Gereffi, Gary [52] is not effective, and most of the backward economies along this path have not achieved high-end upgrading as expected [50].

On the other hand, Economic practice has also proved: that innovation can achieve catch-up development [54], which can be further divided into breakthrough, asymptotic and improved innovation [54]. In the context of reengineering, Tan Renhe et al. believes the positive effect of technological innovation is an important opportunity to achieve high-end in the locked countries[48].(line 194-219)

Added references as following:

  1. 39. The Global Industry Research Institute of Qinghua University.Research Report on the Digital Transformation of Chinese Enterprises (2020). Qinghua University Press.2021.08(in chinese)
  2. 40. Hui Zhibin, Xu Limei, Wang Yingbo.Global Digital Economy Competitiveness Development Report (2020).Social Sciences Literature Press..ISBN:978-75201-7550-0
  3. W B He .Analysisonthe Effect of Digitalization Promoting the High End of Chinas Manufacturing Value Chain.Economic managementOF East China.2020(12):29-37.DOI:10.13546/j.cnki.tjyjc.2021.10.021

  1. KAPLINSKY R. Spreading the Gains from Globaliaztion:Whatean Belearned from Valuechain Analysis?. Jour nal ofDevelopment Studies, 2000, 37 (2): 117-146 . DOI: 10.1080/713600071
  2. G Yoguel.Carlota Perez, Luc Soete - Catching up in technology: entry barriers and windows of opportunity. Revista Brasileira de Inovação, 2015 14(2):257, DOI: 10.20396/rbi.v14i2.8649108
  3. T Amoako;HS Zhang;CSK Dogbe;WWK Pomegbe.; 2020. Effect of internal integration on SMEs’ performance: the role of external integration and ICT. 2020(12). DOI: 10.1108/IJPPM-03-2020-0120
  4. BELLGG. Cluster, Networks and Firm Innovernativeness Strategic Management Journal, 2005, 26: 287-295. DOI: 10.1002/smj.448

50. Zhang Huiqing, Zhai Xiaoqiang.The Characteristics and Enlightenments of China's Participation in Global Value Chains..The Journal ofQuantitative & Technical  Economics.2018(1).DOI:http://en.cnki.com.cn/ /CJFDTotal-SLJY201801001.htm  (in Chinese)

  1. G Yoguel.Carlota Perez, Luc Soete - Catching up in technology: entry barriers and windows of opportunity.Revista Brasileira de Inovação. 14(2): 257,2015. DOI:10.20396/rbi.v14i2.8649108
  2. Gereffi, Gary,Lee, Joonkoo.Economic and Social Upgrading in Global Value Chains and Industrial Clusters: Why Governance Matters. Journal of Business 2016 (133): 25–38  DOI: 10.1007/s10551-014-2373-7
  3. CHRISTENSEN C M. What is disruptive innovation?Harvard Business Review, 2015, 93(12) 44-53.
  4. Gina Colarelli O'Connor, Richard DeMartino.Organizing for Radical Innovation: An Exploratory Study of the Structural Aspects of RI Management Systems in Large FirmsEstablished. Journal of Product Innovation Management,2006, 23 (6): 475-497. DOI: 10.1111/j.1540-5885.2006.00219.x
  5. Y H Zhang.The impact of the global spread on the supply chain of the industrial value chain.The Northern Economy. 2020(5) :1-3.DOI:CNKI:SUN:BFJJ.0.2020-05-002( in chinese)
  6. Manyika, J.; Chui, M.; Brown, B.; Bughin, J.; Dobbs R.; Roxburgh C.; Byers A.H. Big data: the next frontier for innovation, competition, and productivity. McKinsey Global Institute2011, 156.

  1. Results should be discussed.

Response 5:We made a lot of revisions to this section. Please see Line 651-736, Page 20-21.

4.5 Results Description

4.5.1 Content of the study results  analysis

The dominant competition and strategic layout of major countries in the digital economy are increasingly fierce.Major European and American countries in the process of manufacturing digitization,It is common to promote the key application of production process control and explore the potential of data,, its digital level, high quality, leading to its manufacturing global value chain forward participation, high research and development investment, innovation ability, it more in the middle of the global value chain division of labor, with strong international competitiveness.However, due to the unbalanced digital development, weak new infrastructure construction and poor innovation capacity, On the high-end path dependence,China's manufacturing industry should drive the transformation and upgrading of the value chain with digital technology(44)

.4.5.2 Analysis results of manufacturing global value chain participation model

The domestic added value (DVA) eventually absorbed abroad in China's manufacturing industry is rising, The transformation and upgrading of the manufacturing industry was accelerating; The added value of manufacturing products being first exported to abroad and then returned to final consumption in China is relatively low,, China is "made in the world.", not "created in the world"; China's export added value derived from foreign countries increased  about 40%, It shows that the external dependence is improved in the process of China's manufacturing industry digital transformation; Cross-border links of China's manufacturing industry chain have been lengthened, At the lower end of the "smile curve". under the combined impact of increased global production and operating costs, economic recession and enhanced trade protection, Development is even more difficult.(45)

4.5.3 Forward and backward participation analysis results of Chinese GVC

In 2000~2017, GVC forward participation in most Chinese manufacturing industries has increased, backward participation in the process of participating in GVC division of labor, is reducing ;In the process of participating in the GVC division of labor, China is reducing its dependence on foreign added value, and exporting domestic added value,, The network participation mode of some middle and high-end manufacturing departments has gradually shifted from "bottom embedding" to "high-level penetration".

4.5.4 Model test results of the benchmark model and the added interaction term

The variables of digital, foreign demand and factor productivity have obvious positive effect. the marginal effect of digital economy on labor-intensive industries is not significant.

4.6 the existing body of knowledge and Methodological Limitation

4.6.1 Innovation points

Firstly,This paper uses the theory of international competitive advantage to analyze the influence of digitalization on the division of labor status of the GVC in the manufacturing industry, From the micro level, the mechanism of digitalization affects the production mode, innovation mode, organizational form and business mode of manufacturing enterprises, thus it enhance the division of labor status of manufacturing enterprises..

Secondly, the digital index is used for quantitative analysis.In previous studies, few quantitative analysis of the impact of digital on manufacturing competitiveness and global value chain status. This paper uses the index method of ICT in manufacturing added value,

which reflects the digital level of manufacturing , its establishment structure equation, analyzes the influence on the status division of global value chain (GVC) in manufacturing industry. makes innovations in theoretical and quantitative analysis methods.In addition, in terms of database application, combining WIOD data with ADB database , it ,solves the problems he time-series data span and the new data application  in  the export added value of the manufacturing industry.

4.6.2 Contradicting

In the model test of the benchmark model and the added interaction terms (Table 9),, The digital and research and development investment were verified with GVC,R & D investment (rede) did not find a significant utility here, which is clearly against general expectations,. It is reasonable speculation that the knowledge and technology density in the 18 departments shows great heterogeneity and instability, resulting in no significant effect of R & D investment on the improvement of the value chain.

4.6.3 confirming results

By observing the proportion of international trade added value of major manufacturing countries in the main sector, we found  developed countries occupy the research, production and marketing link of core equipment and key parts, so their average GVC position and competitiveness are better than developing countries.Therefore, it is concluded :China's manufacturing industry participating in the global value chain competition, it is faced with the risk of "the lack of high-end innovation capacity leads to the lack of core competitiveness ,low-end production costs increased ".Therefore, we should increase investment in research and development, improve our innovation capacity, increase reform and opening up, actively participate in the division of labor of the global value chain, and enhance the core competitiveness of China's manufacturing industry.

4.6.4 Limitations, outlook of the research

The quantitative determination method of digital level needs to be further improved; strengthen the comprehensive application of the latest RECD database to improve the evaluation accuracy of the impact of digital economy on the global value chain of manufacturing industry.

In the future, this paper will strengthen the research onthe logical mechanism of the impact of the digital industry on the global value chain improvement of the manufacturing industry and the supporting index system of the digital level determination, and further analyze the impact of the digital economy on the status and length of the global value chain of the manufacturing industry.

Enhance the comprehensive use of WIOD, ADB, and RECD databases, and increase the number of world sample analysis countries, in order to improve the accuracy and representativeness of the research results.

  1. Research limitations and outlook should be given.

Response 6:Please see Line 724-736, Page 22.

4.6.4 Limitations、outlook of the research

The quantitative determination method of digital level needs to be further improved; strengthen the comprehensive application of the latest RECD database to improve the evaluation accuracy of the impact of digital economy on the global value chain of manufacturing industry.

In the future, this paper will strengthen the research on the logical mechanism of the impact of the digital industry on the global value chain improvement of the manufacturing industry and the supporting index system of the digital level determination, and further analyze the impact of the digital economy on the status and length of the global value chain of the manufacturing industry.

Enhance the comprehensive use of WIOD, ADB, and RECD databases, and increase the number of world sample analysis countries, in order to improve the accuracy and representativeness of the research results.

Round 2

Reviewer 1 Report

Dear authors, thanks for working on the comments and submitting a revised version. The paper has improved in some parts but got more unstructured at the later parts (section 4 and 5).

Also, I am still missing a clearer elaboration of the research aims and questions of the paper. In the now revised version, you added an additional section for this. Still, the formulation are not clear, also due to some grammar mistakes. For example, take this sentence: "On the bases of existing literature research, the gap for a scientific definition of the digital economy, quantitative analysis, the application of scientific division of labor of manufacturing global value chain, and research on how to accelerate the competitiveness of the global value chain of China's manufacturing chain." --> there is no predicate. Is this the gap you address?

Also, at this early section of the paper, i would rather stick with an argumentation line based on research aims and how you will reach them, then with already discussion the results here (this has to be placed in the discussion section).

The result section section needs big improvement - i would add a separate section "5. Discussion of Results" (instead of 4.5.), followed by section "6. Conclusion and Outlook" (which is called "Revelations" - that term is imho not suitable in a paper). Currently, it is not clear, where results are shown and where implications of these are discussed. Also, currently, there is no outlook for future research for academics (for policy making of course there is). The paper need significant improvement in this parts.

Some points:

  • You don't have to add separate level-3 headings "4.6.2 Contradicting" and "Confirming" - please just discuss it textually in the discussion section.
  • You now state some interesting expected and unexpected findings but you are not elaborating on their implications in detail - what is their impact on your research aim? What is the impact on practice and academia? It is really needed to discuss results in more detail - this is were new knowledge is created and what forms the basis for future research.
  • Conclusion section is clear for policy making, but should be also cleat for academics - for this, a reflection of research gaps / aims and a derivation of future research needs is needed. Also, i would move the limitation section from 4.6.4. to the conclusion section.

In summary, your paper is interesting and the results produced are good, however, you have to work on how to put them in the context of academia in more detail. The research gap and discussion section as well as the conclusion and outlook section still require revision. The current structure in sections 4, 5 and 6 makes it hard to follow and grasp the contribution of the paper, please work on that, the results are there and just have to be presented in a clearer way. I suggest to work on these sections and also close the line argumentation started in the research gap section in the discussion and conclusion section (improve the red-line throughout the paper).

Author Response

Response to the Reviewer’s Comments

Ref No.: sustainability-1593970

Title: Research on the Effect of the Promotion of China's Manufacturing Global Value Chain Driven by the Digital Economy

We thank you for your detailed comments and valuable suggestions on our paper. We have carefully considered and fully addressed all the comments raised by the reviewers in this revision. One of the authors is a native speaker of the English language, and she is responsible for the language revision and polishing of the whole paper. Our point-to-point response to your comments is presented as follows.

  1. The paper has improved in some parts but got more unstructured at the later parts (section 4 and 5).

Response 1:

Parts 4 and 5 of this paper have been revised, and the full text is divided into six parts. Please see Line 651-829.

  1. Also, I am still missing a clearer elaboration of the research aims and questions of the paper. In the now revised version, you added an additional section for this. Also, Still, the formulation are not clear, also due to some grammar mistakes. For example, take this sentence: "On the bases of existing literature research, the gap for a scientific definition of the digital economy, quantitative analysis, the application of scientific division of labor of manufacturing global value chain, and research on how to accelerate the competitiveness of the global value chain of China's manufacturing chain." --> there is no predicate. Is this the gap you address?

Response 2:

Please see Line 280-295

Based on the limitations of existing literatures, this paper attempts to fill the gaps regarding: (1) defining the scientific implications of the digital economy,and analyzing the driving mechanism of the digital economy to promote the upgrading of manufacturing global value chain [6].(2) using the digitalization index(DMI) to analyze the overall and sub-industry digital level of manufacturing industries in China and major countries, and analyze the manufacturing competitiveness to make up for the gap of quantitative assessment on digital technology in literature [7]; (3) using the WIOD input and output table, and production added value decomposition model (WWYZ) to compare analysis of the status and participation index of the global value chain in major manufacturing countries by studying the vertical specialization of manufacturing and global location in value chains in major countries to make up the gap in trade added methods of the research literature [17,18]; (4) analyzing the global value chain ‘s participation degree and characteristics from the dual perspective of forward connection and backward connection to make up the gap of a single perspective to existing literature research [21,22]; (5) analyzing the rising effect of digitalization on the manufacturing global value chain. [24]

  1. Also, at this early section of the paper, i would rather stick with an argumentation line based on research aims and how you will reach them, then with already discussion the results here (this has to be placed in the discussion section).The result section needs big improvement - i would add a separate section "5. Discussion of Results" (instead of 4.5.)

Response 3:

Please see Line 644-725

  1. Results of Discussion

5.1 The digital advantages of China's manufacturing industry are still not obvious

The results of the digital index study show that the total scale of global digital economy development reached 30.2 trillion US dollars, accounting for 40.3% of global GDP in 2020. [40] The United States ranks first with 12.34 trillion US dollars, and China has 4.73 trillion US dollars.

Although the scale of China's digital economy is already at the forefront of the world, the development of the digitally enabled manufacturing industry is still lagging. New infrastructure is weak and has poor innovation capacity. In the high-end path, China's manufacturing industry should drive the transformation and upgrading of the value chain with digital technology.[51] Major countries in Europe and the United States are in the process of manufacturing digitization. They promoted the production process control through an industrial Internet of Things platform. Data technology is more commonly used in the manufacturing industry and its digital level and quality are high. This leads to high forward participation in the global value chain of its manufacturing industry, high R&D investment, and strong innovation ability. Therefore, most of them are in the middle and high end of the global value chain division of labour and have strong international competitiveness.

This paper also found that digitalization has a differentiated effect on the global value chain of China's manufacturing sector. From the perspective of labour-intensive and capital-intensive manufacturing sectors, digital investment increases by 1%, and global value chain participation increases by 0.027% and 0.038%, respectively. Sectors with high R&D capabilities have obvious effectiveness, which indicates that high-tech manufacturing sectors have more competitive advantages, and the effect of digital integration is the most obvious. These industries that have mastered core technologies should construct Chinese own brands and standards and strengthen the international cooperation, [52]which helps break down barriers to embedding in high-end markets.[53]

5.2. The research results of the global value chain participation index and location in the manufacturing industry

Firstly, China's manufacturing industry is located at the low end of the global value chain, but its upgrading is accelerating. The domestic added value is ultimately absorbed abroad. Among them DVA in China's manufacturing industry is rising. It indicates that China's manufacturing industry participation in the division of global value chain is increasing. The added value (RDV) of manufacturing products exported abroad and returned to China is relatively low. China is "made in the world", not "created in the world"; China's foreign added value in domestic exports(FVA) increased about 40%. It shows that the external dependence in China's manufacturing industry's digital transformation was enhanced. Cross-border link extension of China's manufacturing industry chain (PDC value increase). The development is even more difficult under the impact of the higher global production, the operating costs, the economic recession, and the increased trade protection.

Secondly, this paper analyzes the manufacturing global value chain location and participation in the sample countries. The results show that the manufacturing GVC status index of France, the United Kingdom, Estonia and Mexico maintained a continuous downward trend in the previous years from 2000-2017, indicating that the international division of labour status of the manufacturing industry in these countries continued to decline. The manufacturing GVC status index of Canada and Sweden also showed a downward trend. As export-oriented countries, these countries have a relatively high degree of economic dependence on foreign countries and are more vulnerable to the current global economic downturn. The division of labour in the global value chain of Germany, Denmark, South Korea, Japan, and the United States has improved, and the GVC status index has shown an upward trend in fluctuations.

5.3.The forward and backward participation index of Chinese GVC results

In 2000~2017, GVC forward participation in most Chinese manufacturing industries has increased. . In the process of participating in the GVC division of labour, China is reducing its dependence on foreign added value and exporting domestic added value. The network participation mode of some middle and high-end manufacturing departments has gradually shifted from "bottom embedding" to "high-level penetration".

5.4 Model test results of the benchmark model and the added interaction term

The variables of digital, foreign demand and factor productivity have an obvious positive effect. The marginal effect of the digital economy on labor-intensive industries is not significant[52].

5.5 Results of confirmed and Contradicting

By observing the proportion of international trade added value of major manufacturing countries in the main sector, we found that the developed countries occupy the R&D, production, and marketing link of core equipment and key parts. Their average GVC position and competitiveness are better than developing countries. Therefore, China's manufacturing industry participating in the global value chain competition has been faced with the risk of "the lack of high-end innovation capacity leading to the lack of core competitiveness, and low-end production’s increasing costs". So, China should increase the investment in R&D, improve its innovation capacity, increase reforming and opening up, actively participate in the division of labour of the global value chain, and enhance the core competitiveness of China's manufacturing industry.

However, the research results also show certain contradictions. In the test of the benchmark model and the added interaction terms (Table 10), the R & D investment (rede) has no significant positive impact on the rising status of GVC. This is clearly going against general expectations. Reasonable speculation is that the knowledge and technology density showed great heterogeneity in the 18 departments, resulting in no significant effect of rede on the rising status of GVC.

  1. Followed by section "6. Conclusion and Outlook" (which is called "Revelations" - that term is imho not suitable in a paper)

Response 4:

This paper has been modified. Please see Line 726-806

  1. Conclusion, Outlook and suggestion

6.1. Conclusion

In the context of a new round of scientific and technological revolution and the deep adjustment of the international division of labour patterns, the digital economy plays an increasingly important role in improving the GVC status of the manufacturing industry and promoting industrial transformation and upgrading.

They are the priorities and hotspots of the current research that effectively measure the division of labor of China's manufacturing industry in the GVC and its impact of digitization. By establishing an accounting framework for production decomposition models, this paper completely and accurately describes the position and division of labour of China's manufacturing industry in the GVC. At the same time, it uses the digital index to analyze the impact on the upgrading of the GVC and uses the digital index to measure the impact of digitization on the upgrading of the manufacturing GVC. The contribution of the paper is mainly reflected in the following parts.

6.1.1. Defining the concept of digital economy and measure the digital level of manufacturing industry by the digital index (DMI).

This paper analyzes the impact of digitalization on manufacturing industry in China and major sample countries from 2005 to 2020 using data from input-output tables. From 2005 to 2017, China, South Korea, Japan, the United States, Mexico, Finland and other countries had high levels of digitization. In terms of industrial digitization, ICT added value and industrial added value accounted for more than 1/3 in South Korea, Germany, the United States, Ireland, the United Kingdom, and Japan in 2019, with South Korea accounting for 45%, the highest. China's industrial digitalization accounts for only 18.3% of the added value of the industry, which is obviously in the initial stage of digital transformation.

6.1.2. The improved production decomposition model (WWYZ) is established for the first timeUnder the trend of more vertical division of labor in world trade, this paper establishes a production decomposition model (WWYZ) using the latest data of the world input-output table (WIOD) and ADB database from 2000 to 2017 and revises the original export trade decomposition model. This paper extends the analysis of manufacturing GVCS from the export trade stage to the production stage, decomposes the direction of added value created by domestic manufacturing production (forward link) and the source of added value used by domestic production (backward link). It also measures the division of labor status of China's manufacturing industry in the GVC. The paper also analyzed 17 major countries manufacturing GVC participation position, division of labour, etc. These studies make up for the gap that the original study only considered the export trade and ignored the domestic demand .

Therefore, from the overall GVCS index of the manufacturing industry, developed economies such as the United States, Europe, and Japan are still occupying the core leading position in GVC. Although China's manufacturing industry ranks at the forefront of the GVC division of labour among the sample countries, its backward participation is relatively high, and its industrial chain has a long transnational link. This shows that China's manufacturing industry is still embedded in the GVC from the lower level, and its competitiveness is not strong.

6.1.3. Digitalization has a significant positive effect on the rise of the global value chain.

With the data of WIOD and ADB in 2016, this paper introduces digitalization index, R&D investment, overseas demand, industry scale, productivity, and other variables, using the benchmark model to test and draw the following conclusions: digitalization is significantly positively correlated with the GVC division of labor status in manufacturing industry.

Digitization has a differentiated effect on the upgrading utility of the three groups' value chain division status. In the labour-intensive or resource-intensive group with low technology density, the upgrading of GVCS in this group showed a negative effect. It shows that the marginal effect of digitalization on these sectors is not significant, and the increase of digitalization has largely suppressed the advantage of low labour costs in China's manufacturing industry. Digitization has a significant positive effect on the upgrading of GVCS in the other two groups, and the middle-high group is stronger than the middle-low group.

The results show that these two groups can effectively improve their embeddedness in GVCS by accelerating the digitization process. In addition, in the interaction term model of digitization and R&D investment, digitization has a negative effect and a positive effect on the 1% significance level for the medium-high knowledge intensity group. It is reasonable to assume that R&D investment is mostly directed to ICT in technology-intensive manufacturing. This suggests that the intensification of R&D investment is conducive to amplifying this effect. Industry scale and foreign demand have a significant positive impact on the overall position of the manufacturing industry and the improvement of GVCS of each sub-industry.

Because the quantitative analysis method of number level has certain limitations, the analysis method needs to be further improved.

6.2 Outlook

In the future, the authors of this paper will strengthen the research on the multi-dimensional evaluation index system of digital economy. Future research will further examine the digital infrastructure, R&D input, talents, technology, digital technology, and digital governance indicators. The authors will further research digital technology's effect on China’s labour-intensive, capital-intensive, and technology-intensive manufacturing industry, and focus on the effect of value creation of the GVC each link of the space network layout. The future research will also involve deepening the research on the integration of digital technology and manufacturing R&D investment and the effect of globalization. Future research will also involve realizing "cross-border links" in the global value chain of China's manufacturing industry under the international vertical division of labor system.

5.Currently, it is not clear, where results are shown and where implications of these are discussed.

Response 5:

The paper has been modified, see 5. Discussion of the results in lines 644-725

  1. Also, currently, there is no outlook for future research for academics (for policy making of course there is). The paper need significant improvement in this parts.

Response 6

Please see Line 795-806

6.2 Outlook

In the future, the authors of this paper will strengthen the research on the multi-dimensional evaluation index system of digital economy. Future research will further examine the digital infrastructure, R&D input, talents, technology, digital technology, and digital governance indicators. The authors will further research digital technology's effect on China’s labour-intensive, capital-intensive, and technology-intensive manufacturing industry, and focus on the effect of value creation of the GVC each link of the space network layout. The future research will also involve deepening the research on the integration of digital technology and manufacturing R&D investment and the effect of globalization. Future research will also involve realizing "cross-border links" in the global value chain of China's manufacturing industry under the international vertical division of labor system.

Some points:

  1. You don't have to add separate level-3 headings "4.6.2 Contradicting" and "Confirming" - please just discuss it textually in the discussion section.

Response 7:

Modifications have been made. Please see Line 708-725

5.5 Results of confirmed and Contradicting

By observing the proportion of international trade added value of major manufacturing countries in the main sector, we found that the developed countries occupy the R&D, production, and marketing link of core equipment and key parts. Their average GVC position and competitiveness are better than developing countries. Therefore, China's manufacturing industry participating in the global value chain competition has been faced with the risk of "the lack of high-end innovation capacity leading to the lack of core competitiveness, and low-end production’s increasing costs". So, China should increase the investment in R&D, improve its innovation capacity, increase reforming and opening up, actively participate in the division of labour of the global value chain, and enhance the core competitiveness of China's manufacturing industry.

However, the research results also show certain contradictions. In the test of the benchmark model and the added interaction terms (Table 10), the R & D investment (rede) has no significant positive impact on the rising status of GVC. This is clearly going against general expectations. Reasonable speculation is that the knowledge and technology density showed great heterogeneity in the 18 departments, resulting in no significant effect of rede  on  the rising status of GVC.

  1. You now state some interesting expected and unexpected findings but you are not elaborating on their implications in detail - what is their impact on your research aim?.What is the impact on practice and academia?

Response 8

Please see Line726-794

  1. Conclusion, Outlook, and suggestion

6.1. Conclusion

In the context of a new round of scientific and technological revolution and the deep adjustment of the international division of labour patterns, the digital economy plays an increasingly important role in improving the GVC status of the manufacturing industry and promoting industrial transformation and upgrading.

They are the priorities and hotspots of the current research that effectively measure the division of labor of China's manufacturing industry in the GVC and its impact of digitization. By establishing an accounting framework for production decomposition models, this paper completely and accurately describes the position and division of labour of China's manufacturing industry in the GVC. At the same time, it uses the digital index to analyze the impact on the upgrading of the GVC and uses the digital index to measure the impact of digitization on the upgrading of the manufacturing GVC. The contribution of the paper is mainly reflected in the following parts.

6.1.1. Defining the concept of digital economy and measure the digital level of manufacturing industry by the digital index (DMI).

This paper analyzes the impact of digitalization on manufacturing industry in China and major sample countries from 2005 to 2020 using data from input-output tables. From 2005 to 2017, China, South Korea, Japan, the United States, Mexico, Finland and other countries had high levels of digitization. In terms of industrial digitization, ICT added value and industrial added value accounted for more than 1/3 in South Korea, Germany, the United States, Ireland, the United Kingdom, and Japan in 2019, with South Korea accounting for 45%, the highest. China's industrial digitalization accounts for only 18.3% of the added value of the industry, which is obviously in the initial stage of digital transformation.

6.1.2. The improved production decomposition model (WWYZ) is established for the first timeUnder the trend of more vertical division of labor in world trade, this paper establishes a production decomposition model (WWYZ) using the latest data of the world input-output table (WIOD) and ADB database from 2000 to 2017 and revises the original export trade decomposition model. This paper extends the analysis of manufacturing GVCS from the export trade stage to the production stage, decomposes the direction of added value created by domestic manufacturing production (forward link) and the source of added value used by domestic production (backward link). It also measures the division of labor status of China's manufacturing industry in the GVC. The paper also analyzed 17 major countries manufacturing GVC participation position, division of labour, etc. These studies make up for the gap that the original study only considered the export trade and ignored the domestic demand .

Therefore, from the overall GVCS index of the manufacturing industry, developed economies such as the United States, Europe, and Japan are still occupying the core leading position in GVC. Although China's manufacturing industry ranks at the forefront of the GVC division of labour among the sample countries, its backward participation is relatively high, and its industrial chain has a long transnational link. This shows that China's manufacturing industry is still embedded in the GVC from the lower level, and its competitiveness is not strong.

6.1.3. Digitalization has a significant positive effect on the rise of the global value chain.

With the data of WIOD and ADB in 2016, this paper introduces digitalization index, R&D investment, overseas demand, industry scale, productivity, and other variables, using the benchmark model to test and draw the following conclusions: digitalization is significantly positively correlated with the GVC division of labor status in manufacturing industry.

Digitization has a differentiated effect on the upgrading utility of the three groups' value chain division status. In the labour-intensive or resource-intensive group with low technology density, the upgrading of GVCS in this group showed a negative effect. It shows that the marginal effect of digitalization on these sectors is not significant, and the increase of digitalization has largely suppressed the advantage of low labour costs in China's manufacturing industry. Digitization has a significant positive effect on the upgrading of GVCS in the other two groups, and the middle-high group is stronger than the middle-low group.

The results show that these two groups can effectively improve their embeddedness in GVCS by accelerating the digitization process. In addition, in the interaction term model of digitization and R&D investment, digitization has a negative effect and a positive effect on the 1% significance level for the medium-high knowledge intensity group. It is reasonable to assume that R&D investment is mostly directed to ICT in technology-intensive manufacturing. This suggests that the intensification of R&D investment is conducive to amplifying this effect. Industry scale and foreign demand have a significant positive impact on the overall position of the manufacturing industry and the improvement of GVCS of each sub-industry.

Because the quantitative analysis method of number level has certain limitations, the analysis method needs to be further improved.

  1. It is really needed to discuss results in more detail - this is were new knowledge is created and what forms the basis for future research;Conclusion section is clear for policy making, but should be also cleat for academics - for this, a reflection of research gaps / aims and a derivation of future research needs is needed.

Response 9:

The paper has been revised. Please see the Line795-806

6.2 Outlook

In the future, the authors of this paper will strengthen the research on the multi-dimensional evaluation index system of digital economy. Future research will further examine the digital infrastructure, R&D input, talents, technology, digital technology, and digital governance indicators. The authors will further research digital technology's effect on China’s labour-intensive, capital-intensive, and technology-intensive manufacturing industry, and focus on the effect of value creation of the GVC each link of the space network layout. The future research will also involve deepening the research on the integration of digital technology and manufacturing R&D investment and the effect of globalization. Future research will also involve realizing "cross-border links" in the global value chain of China's manufacturing industry under the international vertical division of labor system.

  1. Also, i would move the limitation section from 4.6.4. to the conclusion section.

Response 10:

Please see Line793-794

Because the quantitative analysis method of number level has certain limitations, the analysis method needs to be further improved.

  1. In summary, your paper is interesting and the results produced are good, however, you have to work on how to put them in the context of academia in more detail.

Response 11:

Please see Line795-806

6.2 Outlook

In the future, the authors of this paper will strengthen the research on the multi-dimensional evaluation index system of digital economy. Future research will further examine the digital infrastructure, R&D input, talents, technology, digital technology, and digital governance indicators. The authors will further research digital technology's effect on China’s labour-intensive, capital-intensive, and technology-intensive manufacturing industry, and focus on the effect of value creation of the GVC each link of the space network layout. The future research will also involve deepening the research on the integration of digital technology and manufacturing R&D investment and the effect of globalization. Future research will also involve realizing "cross-border links" in the global value chain of China's manufacturing industry under the international vertical division of labor system.

  1. The research gap and discussion section as well as the conclusion and outlook section still require revision. The current structure in sections 4, 5 and 6 makes it hard to follow and grasp the contribution of the paper

Response 12:

Please see Line 644-864:

  1. Results of Discussion----6.0.Conclusion,Outlook and suggestion..........
  2. Please work on that, the results are there and just have to be presented in a clearer way.

Response 13:

The paper has been revised,Please see Line 644-725:

  1. Results of Discussion
  2. I suggest to work on these sections and also close the line argumentation,started in the research gap section in the discussion and conclusion section (improve the red-line throughout the paper).

Response 12:

The paper has been revised. Please see Lline:279-295: 

2.3.5 Research Aim, Research Questions, and Making up the gap;

Line:740-792:

6.1.1Defining the concept of digital economy and measure the digital level of manufacturing industry by the digital index (DMI)—6.1.3. Digitalization has a significant positive effect on the rise of the global value chain.

Round 3

Reviewer 1 Report

You did several changes in the paper and added new parts as well. Unfortunately, the most important comments of my review have still not been sufficiently addressed after two rounds. The paper is still missing clear research objectives (it just states that some gaps are addressed, but not what the results / aims of the paper are. The results and discussions section got unfortunately even more complex and harder to follow. Headings like "Results of Discussion" (do you mean discussion of results?) or "5.5 Results of confirmed and Contradicting" are confusing. 

Unfortunately, i don't see a high chance, that the paper would improve after a third revision up to the point which meets academic rigor and the quality standards of this journal. However, i suggest submitting to a different outlet or better to a conference with a clearer focus in mind (i.e., reduce complexity by picking out smaller deliveries).